# QUANTILE-FREE REGRESSION: A FLEXIBLE ALTERNATIVE TO QUANTILE REGRESSION

## ABSTRACT

Constructing valid prediction intervals rather than point estimates is a well-established method for uncertainty quantification in the regression setting. Models equipped with this capacity output an interval of values in which the ground truth target will fall with some prespecified probability. This is an essential requirement in many real-world applications in which simple point predictions' inability to convey the magnitude and frequency of errors renders them insufficient for high-stakes decisions. Quantile regression is well-established as a leading approach for obtaining such intervals via the empirical estimation of quantiles in the (non-parametric) distribution of outputs. This method is simple, computationally inexpensive, interpretable, assumption-free, and highly effective. However, it does require that the quantiles being learned are chosen a priori. This results in either (a) intervals that are arbitrarily symmetric around the median which is sub-optimal for realistic skewed distributions or (b) learning an excessive number of intervals. In this work, we propose Quantile-Free Regression (QFR), a direct replacement for quantile regression based interval construction which liberates it from this limitation whilst maintaining its strengths. We demonstrate that this added flexibility results in intervals with an improvement in desirable qualities (e.g. sharpness) whilst retaining the essential coverage guarantees of quantile regression.

## 1 INTRODUCTION

Reliable uncertainty estimation is an essential requirement for safely and robustly deploying neural networks in real-world applications (Amodei et al., 2016; Dietterich, 2017; Kompa et al., 2021). However, research has consistently shown this to be a challenging problem in practice (Guo et al., 2017; Yao et al., 2019; Ayhan & Berens, 2022). Therefore, significant efforts have been made to address this task in order to contribute towards more reliable and trustworthy models (see e.g. Gawlikowski et al. (2023)). A significant aspect of this effort is developing regression methods that output predictive intervals rather than point predictions. This has proven to be a crucial requirement in high-stakes applications including medical decision-making (Begoli et al., 2019), autonomous driving (Su et al., 2023), and energy forecasting (Wang et al., 2022).

Especially in the case of neural networks, *quantile regression* (Koenker & Bassett Jr, 1978) has emerged as a powerful method for obtaining such intervals. This approach requires the model to output estimates of two quantiles rather than a single point prediction which is easily optimized in practice by a simple change in loss function. These quantiles may then be used to construct an interval $(\mu_1, \mu_2)$ within which the true label will lie with probability $\alpha$ (a formal description of quantile regression is provided in Section 2). Obtaining predictive intervals via quantile regression has earned substantial popularity in both research and practice (Koenker & Hallock, 2001; Koenker, 2017; Yu et al., 2003; Fitzenberger et al., 2001). This uptake can be attributed to several factors including (a) methodological simplicity requiring minimal changes to the modeling procedure, (b) negligible increased computational cost (in contrast to e.g. ensemble methods (Lakshminarayanan et al., 2017)), (c) simple, easily interpreted characterization of uncertainty (Savelli & Joslyn, 2013; Goodwin et al., 2010), (d) lack of parametric assumptions on the data-generating process, and (e) enduring empirical effectiveness (Chung et al., 2021; Tagasovska & Lopez-Paz, 2019). Furthermore, quantile regression methods can be wrapped in the conformal prediction procedure of Vovk et al. (2005) to additionally provide finite sample coverage guarantees as demonstrated in Romano et al.

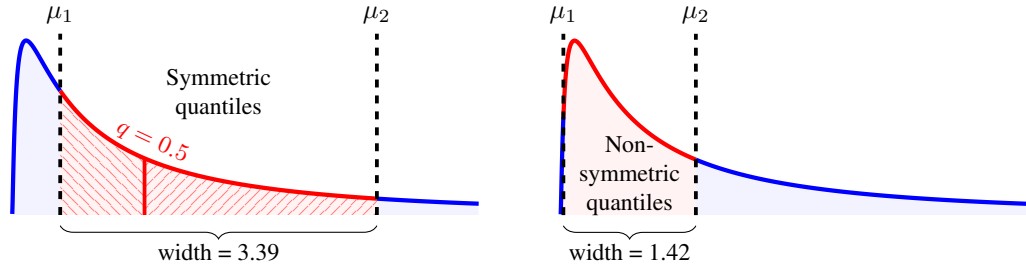

Figure 1: **Symmetric quantiles.** We compare two pairs of intervals on an identical (non-symmetric) log-normal probability distribution where in both cases a fixed level of coverage $\alpha$ is obtained. In the left figure, the intervals are selected to be symmetric in terms of probability mass around the median $q = 0.5$ (i.e. the two lined regions and contain equal probability mass) as in the case of quantile regression. In the right figure, we remove this constraint and obtain a much narrower interval with identical coverage.

(2019). Several variations of quantile regression have been introduced in recent years (see Section 2) which are typically evaluated based on desirable properties of their resulting prediction intervals. These include satisfying the required level of coverage, minimizing interval width, and achieving improved conditional coverage (also see Section 2).

In this work our **contributions** are threefold: **(1)** In Section 3.1 we identify a substantial inefficiency in the standard procedure of first estimating quantiles from which predictive intervals are then derived. We show that this typically results in intervals with their midpoint fixed at the median which is undesirable for non-symmetric distributions (Figure 1) *or* requires learning more quantiles than necessary resulting in a more difficult learning problem and, therefore, sub-optimal performance (see e.g. SQR in Section 2); **(2)** In Section 3.2 we propose a novel objective which directly learns intervals *without* a priori specifying particular quantiles. We then equip this function with a regularization term that aids in selecting among possible interval choices by rewarding desirable properties such as narrower intervals or improved conditional coverage. This results in a method we term Quantile-Free Regression (QFR); **(3)** We theoretically show that the solution of our regularized objective asymptotically achieves valid coverage (Section 3.2). Empirically, we find that it results in superior performance to existing methods when evaluated on standard benchmarks (Section 4).

## 2 BACKGROUND

In this section, we provide a summary of relevant existing works that convert neural networks from outputting point estimates to outputting predictive intervals (i.e. *single model approaches*). In Appendix A we provide a broad summary of predictive interval generation more generally and highlight some of the unique advantages of the single model approach that we consider in this text.

**Deriving intervals from quantiles.** Throughout this work we consider the standard regression task consisting of input/target pairs $(\mathbf{X}, Y) \in \mathbb{R}^d \times \mathbb{R}$ with $d \in \mathbb{N}$ where bold denotes vectors and non-bold denotes scalars. We express realizations of these random variables (i.e. data) using lower-case $(\mathbf{x}, y)$. Denoting the cumulative distribution function of a probability distribution with $\mathbb{F}$, we recall that a *quantile function* is given by $\mathbb{F}^{-1}(p) = \inf\{q \in \mathbb{R} : p \leq \mathbb{F}(q)\}$ where $p \in (0, 1)$ is the desired *quantile probability*. Specifically, this provides some *quantile value* $q$ such that $\mathbb{P}(Y \leq q) = p$. In the machine learning setting, we are generally interested in the probability distribution of $Y$ *conditional* on a given input $\mathbf{X} = \mathbf{x}$. Throughout this work, we refer to the task of estimating a quantile value $q$ corresponding to a particular quantile probability $p$ from data as *estimating quantiles*. Once we have some function $\mu : \mathbb{R}^d \to \mathbb{R}$ for estimating quantiles (e.g. a neural network), we might wish to construct an *interval* such that $\mathbb{P}(\mu_1(\mathbf{x}) \leq Y \leq \mu_2(\mathbf{x})) = \alpha$ with $\mu_1(\mathbf{x}) < \mu_2(\mathbf{x})$[1] and $\alpha \in (0, 1)$. In other words, a pair of bounds between which the target will lie with some desired probability $\alpha$. We will generally drop the dependence of $\mu_1$ and $\mu_2$ on $\mathbf{x}$ for ease of notation. Clearly, this interval can be easily derived from the quantile function by simply noting that $\mathbb{P}(\mu_1 \leq Y \leq \mu_2) = \mathbb{P}(Y \leq \mu_2) - \mathbb{P}(Y \leq \mu_1)$. Therefore the problem of constructing valid intervals may be solved by approximating the quantile function to estimate the appropriate quantiles *and then* constructing an interval from these quantile values. However, this assumes that we know

---

[1]In fact, our proposed method described in Section 3.2 is *permutation invariant* with respect to the two bounds as we discuss in detail in Appendix E.

Table 1: **Quantile regression methods.** Several desirable properties of these methods are considered. Note that $\frac{1}{2}$ denotes partially satisfying a property as the WS, SQR & IR objectives can trade-off between coverage and interval width but do not directly consider conditional coverage.

| Property | QR | WS | SQR | IR | This work |
|---|---|---|---|---|---|
| Suitable for non-centered distributions | ✗ | ✗ | ✓ | ✓ | ✓ |
| Avoids explicitly learning all quantiles | ✓ | ✓ | ✗ | ✓ | ✓ |
| Dynamically controls the trade-off between any desirable objectives | ✗ | $\frac{1}{2}$ | $\frac{1}{2}$ | $\frac{1}{2}$ | ✓ |
| Asymptotic coverage guarantees | ✓ | ✓ | ✓ | ✗ | ✓ |
| No Gaussian approximation or assumption of iid miscoverage of instances | ✓ | ✓ | ✓ | ✗ | ✓ |

which quantile probabilities we should use in advance as any pair of quantile probabilities $p_l < p_u$ such that $p_u - p_l = \alpha$ will result in an interval with $\alpha$ coverage. We discuss the consequences of this fact in Section 3.1.

**Quantile Regression (QR).** Here we refer to such methods that aim to provide intervals by accurately estimating quantiles. In the case of neural networks, the key distinction from standard point estimation methods which predict the expected value $\mathbb{E}(Y|\mathbf{X})$ lies in the choice of loss function. We require a loss function $\mathcal{L} : \cdot \to \mathbb{R}^+$ mapping from a quantile estimate to a scalar loss upon which we can apply gradient descent. Perhaps the most widely known approach is that of the *pinball loss* function (also known as *quantile loss*) of Koenker & Bassett Jr (1978); Steinwart & Christmann (2011). For a quantile estimator $\mu : \mathbb{R}^d \to \mathbb{R}$, the pinball loss expression is

$$\rho_q(\mu, \mathbf{x}, y) = \begin{cases} q(y - \mu(\mathbf{x})) & \text{if } y - \mu(\mathbf{x}) \geq 0 \\ (q-1)(y - \mu(\mathbf{x})) & \text{if } y - \mu(\mathbf{x}) < 0 \, . \end{cases}$$

Then a strategy to construct an interval of targeted coverage level of $1 - \alpha$ involves estimating two specific conditional quantiles, denoted as $q_l$ and $q_u$, where $q_l$ corresponds to the $\frac{\alpha}{2}$ quantile, and $q_u$ corresponds to the $1 - \frac{\alpha}{2}$ quantile. Thus, the loss function optimized by the neural network is given by

$$\mathcal{L}_\alpha^{\text{QR}}((\mu_1, \mu_2), \mathbf{x}, y) = \rho_{\frac{\alpha}{2}}(\mu_1, \mathbf{x}, y) + \rho_{1-\frac{\alpha}{2}}(\mu_2, \mathbf{x}, y).$$

This methodology ensures that the probability of the ground truth target $y$ falling within the interval $[\mu_1, \mu_2]$ is $1-\alpha$, thereby establishing the desired mean coverage. In practice, two particular quantiles are typically predefined and learned using a single neural network with two outputs. We refer to this antecedent approach as *quantile regression (QR)* throughout this work.

**Simultaneous Quantile Regression (SQR).** Rather than predefining two particular quantiles, Tagasovska & Lopez-Paz (2019) propose to learn *all* possible quantiles with a single output model by augmenting the neural network with an additional input for the desired quantile. We express this simultaneous quantile regressor as $\mu_q(\mathbf{x})$. Throughout training the quantile $q$ is stochastically selected from a uniform distribution where any quantile loss function may be applied (e.g. pinball).

**Winkler Score (WS).** As an alternative to the pinball loss objective, Chung et al. (2021) introduce the *Winkler score* (or *interval score*) as a substitute. Expressing the standard indicator function as $\mathbb{I}$, this objective is given by

$$\mathcal{L}_\alpha^{\text{WS}}((\mu_1, \mu_2), \mathbf{x}, y) = (\mu_2 - \mu_1) + \frac{2}{\alpha}(\mu_1 - y)\mathbb{I}_{y<\mu_1} + \frac{2}{\alpha}(y - \mu_2)\mathbb{I}_{y>\mu_2}.$$

The authors show that the minimum of the expectation of this objective is obtained when $\mu_1$ and $\mu_2$ attain the true (symmetric) conditional quantiles. This objective can lead to *sharper* intervals (i.e. more adaptive coverage conditional on a given example $\mathbf{x}$). Just as in the case of SQR, if we wish to avoid selecting quantile values a priori, the authors propose learning all possible quantiles selected using a uniform distribution during training.

**Interval Regression (IR).** Learning the intervals directly without the intermediate step of first learning quantiles is an alternative approach that has emerged somewhat independently of the quantile regression literature. A method proposed in Pearce et al. (2018) achieves best-in-class empirical

performance by introducing a loss function that attempts to balance coverage with interval width. By making the strong assumptions that (a) the cases of miscoverage are iid and (b) batch sizes are sufficiently large for the binomial distribution to be well approximation by a Gaussian, the authors derive the following objective

$$\mathcal{L}_\alpha^{\text{IR}}((\mu_1, \mu_2), \mathbf{x}, y) = \frac{n}{c}(\mu_2 - \mu_1) \cdot \mathbb{I}_{\mu_1 \leq y \leq \mu_2} + \lambda \frac{n}{\alpha(1-\alpha)} \max(0, (1-\alpha) - \frac{c}{n})$$

$$\text{where } c := \sum_{i=1}^{n} \mathbb{I}_{\mu_1 \leq y \leq \mu_2}.$$

Here $\lambda \in \mathbb{R}$ denotes a hyperparameter weighting term and $n$ denotes the batch size. Unfortunately, unlike quantile regression methods, this approach does not achieve asymptotic coverage guarantees.

**Evaluation.** Competing approaches to obtaining prediction intervals are typically compared across a range of desirable properties. As we only have access to a finite dataset, asymptotic coverage is not guaranteed resulting in the need to evaluate *calibration*. Ideally, we would produce intervals that accurately model the *conditional* probability $P(\mu_1(\mathbf{x}) < y < \mu_2(\mathbf{x})|\mathbf{x})$. Of course, in the standard setting, we cannot directly estimate this quantity (Zhao et al., 2020) and instead often consider the *marginal* probability $P(\mu_1(\mathbf{x}) < y < \mu_2(\mathbf{x}))$. We can estimate the marginal calibration on the test set using the Prediction Interval Coverage Probability (PICP) which simply measures the ratio of observations falling inside their intervals (Kuleshov et al., 2018; Tagasovska & Lopez-Paz, 2019).

Despite the aforementioned impossibility of exactly estimating the former conditional quantity, several proxy metrics have been proposed that test for independence between examples $\mathbf{x}$ and instances of miscoverage. These include using Pearson's correlation between interval width and miscoverage cases (Feldman et al., 2021) and the independence promoting Hilbert-Schmidt independence criterion (HSIC) (Greenfeld & Shalit, 2020).

However, probably the most common criteria of evaluation considers aggregate *interval width*. For a fixed level of coverage, narrower intervals are often considered preferable. This can also prevent trivial solutions with some potentially infinite width intervals which may still satisfy empirical tests of coverage. Sometimes referred to as *sharpness* (Gneiting et al., 2007) or *adaptive coverage* (Seedat et al., 2023), we primarily consider Mean Prediction Interval Width (MPIW) which measures the mean interval width across the test data (Tagasovska & Lopez-Paz, 2019). We provide (a) some extended discussion on the motivation of interval width as an objective in Appendix F and (b) a formal description of all evaluation metrics in Appendix D to ensure this work is self-contained.

## 3 A FLEXIBLE ALTERNATIVE TO QUANTILE REGRESSION

### 3.1 HIGHLIGHTING THE LIMITATION OF EXISTING QUANTILE REGRESSION METHODS

Using quantile estimation as a means for obtaining predictive intervals may be viewed as an example of Vapnik's famous heuristic that "when solving a problem of interest, do not solve a more general problem as an intermediate step" (Vapnik, 2006). The fundamental issue becomes apparent by closely considering the quantile regression approach.

The standard quantile regression approach consists of selecting the two quantiles a priori such that the region between them results in a predictive interval. In practice, for a desired coverage level $\alpha$, the standard approach is to select the $\frac{\alpha}{2}$ and $1 - \frac{\alpha}{2}$ quantiles. Indeed, a principled selection of specific non-symmetric quantiles would require knowledge of the underlying noise distribution which is unknown. As illustrated in Figure 1, when the underlying noise distribution around $Y$ is, in fact, non-symmetric this results in wider than necessary intervals due to being arbitrarily centered (in terms of probability mass) around the median[2]. On real-world tasks, we should expect a non-symmetric noise distribution to be ubiquitous. This has been highlighted in previous work (Tagasovska & Lopez-Paz, 2019) and we further illustrate this by including histograms of the target distributions of popular, real-world datasets in Table 3 and their summary statistics in Table 6. Whilst the true noise distribution cannot be known on real data, the shapes of these empirical distributions suggest that perfect symmetry is a very strong assumption to hold over natural phenomena.

---

[2]Of course, if centered intervals are *required* then quantile regression is appropriate. However, this is unlikely to be a wise requirement in the prevalent case of a skewed target distribution.

The existing resolution to this issue, as introduced by Tagasovska & Lopez-Paz (2019), is to learn *all* possible quantile probabilities in $(0, 1)$ (see e.g. SQR in Section 2). This is explicitly aimed at rectifying the aforementioned limitation as the authors note that it enables them to "model non-Gaussian, skewed, asymmetric, multimodal, and heteroskedastic aleatoric noise in data". The idea being that, once all quantiles are learned, any pair may be selected such that they satisfy $\alpha$ coverage in addition to other qualities (e.g. narrower intervals). However, this introduces a significantly more challenging learning problem of estimating an infinite number of quantiles rather than exactly two for each example which can impact the performance of the underlying quantile estimator. In our experiments in Section 4, we show that this approach generally results in poorer performance when compared against the former approach despite its added flexibility.

In Section 3.2 we introduce an alternative approach that solves the interval estimation problem directly. Although the previous work of Pearce et al. (2018) also considers a direct interval regression approach, ours is the first to emerge from the quantile regression literature and thereby is imbued with a notion of distributional quantiles and converges to a solution that achieves asymptotic coverage guarantees. This is reflected in superior coverage when evaluated empirically in Section 4. Table 1 summarizes the key differences between our proposed method and those of previous works.

## 3.2 PROPOSED RESOLUTION: QUANTILE-FREE REGRESSION

**Quantile-Free Regression (QFR).** We begin by introducing a novel objective which *directly* learns intervals without the intermediate step of prespecifying quantiles. For a targeted coverage level $\alpha$ and a neural network outputting two interval bounds $(\mu_1, \mu_2)$ we minimize

$$\mathcal{L}_\alpha^{\text{QFR}}((\mu_1, \mu_2), \mathbf{x}, y) = \begin{cases} \alpha\kappa & \text{if } \kappa \geq 0 \\ (\alpha - 1)\kappa & \text{if } \kappa < 0 \end{cases} \text{ with } \kappa = (y - \mu_1)(y - \mu_2). \tag{1}$$

This expression makes no assumptions about the shape of the target noise distribution. It does not require the interval bounds to be placed at specific quantile values nor does it require the neural network to explicitly model all quantiles. Theorem 3.1 provides formal guarantees that the minimization of this loss function asymptotically yields a valid interval i.e. the coverage rate $\alpha$ is achieved. Note that proofs for all theorems and propositions are provided in Appendix B.

**Theorem 3.1** (QFR Coverage). *For any random variable $Y$ associated with an input $x$, $\forall \alpha \in [0, 1]$,*

$$(\mu_1^*(x), \mu_2^*(x)) = \underset{\mu_1, \mu_2}{\arg\min} \{\mathbb{E}_Y(\mathcal{L}_\alpha^{QFR}((\mu_1, \mu_2), x, Y))\} \implies \mathbb{P}(\mu_1^*(x) < Y < \mu_2^*(x)) = \alpha$$

Additionally, in Theorem 3.2 we show that this objective achieves the correct finite sample coverage when fit to arbitrary data.

**Theorem 3.2** (QFR with finite samples). *For any random variable $Y$ associated with an input $x$, we consider N realizations of this random variable : $\{y_i\}_{i=1,N}$. $\forall \alpha \in [0, 1]$ such that $\alpha \cdot N \in \mathbb{N}$,*

$$(\mu_1^*(x), \mu_2^*(x)) = \underset{\mu_1, \mu_2}{\arg\min} \{\sum_{i=1}^{N} \mathcal{L}_\alpha^{QFR}((\mu_1, \mu_2), x, y_i)\} \implies \frac{1}{N} \sum_{i=1}^{N} \mathbb{I}_{y_i \in [\mu_1*, \mu_2*]} = \alpha$$

**Rewarding preferable solutions.** Given the added flexibility of shifting intervals rather than bounding them around the median, there now exists a potentially infinite number of competing solutions that achieve the desired level of coverage. As the QFR expression is largely agnostic to solutions that obtain additional desirable properties such as narrower interval widths or better conditional coverage we would like to induce some preference among solutions. A natural strategy is to upweight preferable solutions (e.g. narrower intervals or improved conditional coverage) via an additive regularization term $\mathcal{R}$ and a scalar weighting term $\lambda \in \mathbb{R}^+$. Thereby we provide practitioners with the flexibility to choose whichever interval properties provide the most utility. Therefore, the complete QFR-$\mathcal{R}$ (regularized) objective takes the form

$$\mathcal{L}_\alpha^{\text{QFR-R}}((\mu_1, \mu_2), \mathbf{x}, y) = \mathcal{L}_\alpha^{\text{QFR}}((\mu_1, \mu_2), \mathbf{x}, y) + \lambda \cdot \mathcal{R}(\cdot). \tag{2}$$

Given this structure, we now introduce two specific choices for $\mathcal{R}$.

(1) **QFR-W: width minimizing $\mathcal{R}$.** As discussed in Section 2, interval width is a principal criterion for evaluating methods for obtaining predictive intervals (Tagasovska & Lopez-Paz, 2019; Feldman et al., 2021; Romano et al., 2019). A direct approach for minimizing interval width is to penalize the sample-wise squared interval width such that

$$\mathcal{L}_\alpha^{\text{QFR-W}}((\mu_1, \mu_2), \mathbf{x}, y) = \mathcal{L}_\alpha^{\text{QFR}}((\mu_1, \mu_2), \mathbf{x}, y) + \lambda \frac{(\mu_2 - \mu_1)^2}{2}.$$

By integrating this penalty term, we aim to effectively navigate the landscape of potential solutions, encouraging the model to prioritize intervals of reduced length whilst still maintaining the targeted coverage. Analogous to the approach taken in Theorem 3.1, we can extend our analysis to consider this complete loss function. We find that the introduced penalty term induces a bias to the coverage of the interval estimator – the result is now $\mathbb{P}(\mu_1^* < Y < \mu_2^*) = \alpha - 2\lambda$. However, we can easily remove the bias by modifying the targeted coverage rate. By choosing $\hat{\alpha} = \alpha + 2\lambda$, we obtain $\mathbb{P}(\mu_1^* < Y < \mu_2^*) = \hat{\alpha} - 2\lambda = \alpha$

**Theorem 3.3** (QFR-W Coverage). *For any random variable $Y$ associated with an input $x$, $\forall \alpha \in [0, 1]$,*

$$(\mu_1^*(x), \mu_2^*(x)) = \arg\min_{\mu_1, \mu_2} \{\mathbb{E}_Y(\mathcal{L}_{\alpha+2\lambda}^{QFR\text{-}W}(\mu_1, \mu_2), x, Y))\} \implies \mathbb{P}(\mu_1^*(x) < Y < \mu_2^*(x)) = \alpha$$

**Proposition 3.1** (Existence and Uniqueness of Solution). *$\mu_1^{min}$ and $\mu_2^{max}$ denote the boundaries of our optimization problem. For a target distribution $Y$ with a cumulative distribution function that is k-Lipschitz continuous with $k < 1 + \frac{\alpha}{\mu_2^{max} - \mu_1^{min}}$, when $\lambda > max(0, \int_{\mu_1^{min}}^{\mu_2^{max}} d\mathbb{P}_Y(y) - \alpha)$, the minimum of $\mathcal{L}_{\alpha+2\lambda}^{QFR\text{-}W}$ exists and is unique.*

An important additional benefit of the penalty term, presented in Proposition 3.1, is that it makes the loss function convex for a wide range of target distributions and $\lambda$. Hence, leading to a welcome additional result: the existence and uniqueness of its minimum. We later empirically verify (see Section 4) that this objective does perform well on real-world data.

(2) **QFR-O: width-coverage independence $\mathcal{R}$.** As discussed in Section 2, whilst interval construction methods are most commonly evaluated based on their resulting interval width for a realized coverage level (sometimes referred to as the high-quality principle (Pearce et al., 2018)), alternative objectives may also be desirable. Feldman et al. (2021) introduced *orthogonal quantile regression* (OQR) which instead optimized for a notion of *conditional coverage* rather than minimizing interval width. Specifically, the authors introduce a regularization term that promotes independence between the size of the intervals and occurrences of a (mis)coverage event. They combine their proposed regularization term with both QR and IR and report significant gains in measures of conditional coverage. We can easily combine their term with our QFR instead by setting it as the regularization term $\mathcal{R}$ in Equation (2) where

$$\mathcal{R}(\cdot) = \left| \frac{\text{Cov}(\mathbf{w}, \mathbf{m})}{\text{Var}(\mathbf{w})\text{Var}(\mathbf{m})} \right|.$$

With $\mathbf{w}$ denoting the vector of interval widths where $w_i = |\mu_2(\mathbf{x}_i) - \mu_1(\mathbf{x}_i)|$ and $\mathbf{m}$ denoting the indicator vector of coverage events where $m_i = \mathbb{I}_{y_i \in [\mu_1(\mathbf{x}_i), \mu_2(\mathbf{x}_i)]}$ – both calculated on the training data. Thus, this regularization term can simply be interpreted as the Pearson correlation between the interval widths and instances of coverage or miscoverage[3]. We refer to this complete objective as QFR-O (orthogonal) and, as with the other objectives, we provide proof for its asymptotic coverage in Appendix B.

**Trading-off width and orthogonality.** We note that there is typically a trade-off between minimizing interval width and maximizing conditional coverage. As we later observe in Figure 2, obtaining near optimal conditional coverage generally requires wider intervals than is strictly necessary for obtaining valid marginal coverage. We emphasize that in this work we are agnostic as to which qualities are preferable and, instead, we enable the practitioner to make this decision based on their specific application. In the remainder of this paper, we empirically verify that our proposed objective behaves as expected and successfully utilizes its added flexibility to outperform benchmark methods at achieving their respective goals whilst maintaining empirical coverage.

---

[3]Note that taking the absolute value results in this term penalizing correlation between interval width and *either* instances of coverage or miscoverage (as we would desire).

Table 2: **Empirical verification**. Coverage and width ($\pm$ standard errors) are assessed for producing predictive intervals on symmetric and non-symmetric noise distributions. As expected, for realistic, skewed noise distributions, width-minimizing QFR (QFR-W) produces narrower intervals.

| | Coverage | Width | Dist. Histogram |
|---|---|---|---|
| QR | 0.81 ($\pm$0.007) | 0.26 ($\pm$0.003) | |
| QFR (w/o reg) | 0.81 ($\pm$0.003) | 0.26 ($\pm$0.002) | |
| QFR-W (with reg) | 0.82 ($\pm$0.004) | 0.26 ($\pm$0.002) | |
| QR | 0.80 ($\pm$0.002) | 1.61 ($\pm$0.007) | |
| QFR (w/o reg) | 0.80 ($\pm$0.007) | 1.72 ($\pm$0.010) | |
| QFR-W (with reg) | 0.81 ($\pm$0.002) | 1.50 ($\pm$0.005) | |

## 4 EXPERIMENTS

**Empirical verification.** We begin by empirically verifying the predicted gains of the QFR objective over quantile regression on non-symmetric noise distributions. As the noise distribution cannot be known on real-world data, here we generate synthetic data according to a known process. The data is generated according to a data-generating process in which the label is determined according to $Y = X + \epsilon$, where $X \in \mathbb{R}$ represents a deterministic component which is set as constant and $\epsilon \in \mathbb{R}$ is the noise component. Then the noise distribution is selected as either a (symmetric) Gaussian or a (non-symmetric) truncated Gaussian. We fit a simple linear neural network consisting of just a single layer. As illustrated in Table 2, the empirical results match our theoretical expectations. All methods perform equally well on the symmetric Gaussian noise where intervals centered at the median are optimal. However, QR fails to achieve optimal width on the truncated Gaussian due to being arbitrarily centered at the median. Whilst QFR (w/o reg) is unbiased, it is not incentivized to produce narrower intervals and thus performs similarly to QR. Only QFR-W (with reg) achieves the optimally narrow interval solution.

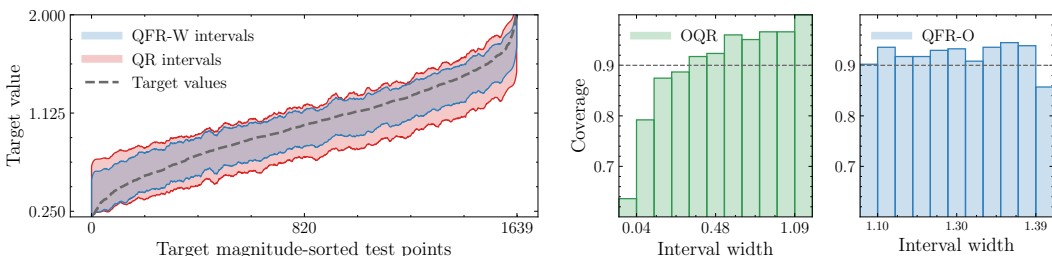

Figure 2: **Applying QFR-W & QFR-O in practice.** Resulting intervals on robotics distance estimation task. **(left)** The QFR-W objective achieves generally narrower intervals across the 1639 test examples. **(right)** The QFR-O achieves more consistent coverage across different interval widths.

**Applying QFR-W & QFR-O in practice.** We now turn our attention to applying our proposed interval prediction approach to practical tasks in high-stakes domains. To illustrate, we consider a task in the robotics domain in which safety and reliability are essential, especially when applied in close proximity to humans (Baek & Kröger, 2023). We consider the task of estimating a robot arm-effector's distance from a target given noisy measurements of inputs such as joint positions or twist angles. The *kin8nm* dataset provides an example of such a task consisting of 8192 measurements (Ghahramani, 1996). For this problem, we desire accurate intervals such that the arm may be used safely and effectively and, depending on the specifics of the application, either interval width or conditional coverage may be important. In Figure 2 we provide the results of applying QFR to this problem using the same experimental setup as described later in the benchmarking experiments.

In the *left-hand side* of this plot, we optimize for interval width using QFR-W and compare the resulting intervals to those using QR. For presentation we sort the test set points according to increasing target magnitude and apply a Savitzky–Golay filter to smooth the intervals (see Figure 4 for a version without smoothing applied). We observe that the QFR-W intervals are noticeably narrower despite providing the same marginal coverage. Then, in the *right-hand side* subplots, we demonstrate the effects of instead optimizing for conditional coverage using QFR-O and evaluate

Table 3: **Benchmarking QFR-W**. We evaluate each method across 9 datasets where the first row reports the marginal coverage obtained and the second row reports interval width as measured using MPIW. All results report the test set mean over 20 runs ($\pm$ a standard error). We ~~strikethrough~~ results where desired coverage is not achieved (see text for details).

| Dataset | Ours | QR | SQR | WS | IR | Dist. Hist. |
|---|---|---|---|---|---|---|
| concrete | 89.95 (0.88) | 89.73 (0.67) | ~~86.49~~ (0.96) | 89.73 (0.75) | 92.16 (0.53) | |
| | **0.43** (0.01) | **0.43** (0.01) | 1.10 (0.09) | 2.42 (0.05) | 0.47 (0.01) | |
| power | 90.73 (0.58) | 92.11 (0.51) | 91.56 (0.35) | 89.96 (0.35) | ~~99.04~~ (0.17) | |
| | **0.03** (0.00) | **0.03** (0.00) | 0.15 (0.02) | 1.86 (0.02) | 0.05 (0.00) | |
| wine | 89.44 (0.72) | 90.91 (0.79) | 87.27 (0.57) | 89.56 (0.98) | 91.05 (0.50) | |
| | **0.35** (0.01) | **0.35** (0.01) | 0.48 (0.02) | 1.95 (0.03) | **0.35** (0.03) | |
| yacht | 93.23 (1.90) | ~~96.37~~ (0.65) | 87.30 (1.10) | 86.85 (2.7) | ~~94.03~~ (0.72) | |
| | **0.27** (0.01) | 0.28 (0.01) | 0.80 (0.05) | 3.20 (0.17) | 0.63 (0.02) | |
| naval | 90.95 (1.90) | 92.11 (1.30) | 92.33 (0.24) | 89.61 (0.37) | ~~100.0~~ (0.00) | |
| | **0.01** (0.00) | 0.02 (0.00) | 0.13 (0.02) | 2.95 (0.03) | 0.04 (0.00) | |
| energy | 89.87 (1.70) | ~~98.31~~ (0.34) | 89.19 (0.70) | 90.19 (0.87) | 93.15 (1.40) | |
| | **0.13** (0.00) | 0.18 (0.01) | 0.44 (0.02) | 2.22 (0.05) | 0.65 (0.08) | |
| boston | 89.80 (0.93) | ~~94.41~~ (0.37) | ~~84.88~~ (0.93) | 87.99 (1.40) | 89.61 (2.40) | |
| | **0.51** (0.01) | 0.58 (0.01) | 0.54 (0.02) | 2.17 (0.03) | 0.55 (0.10) | |
| kin8nm | 90.59 (0.23) | 91.56 (0.41) | 90.89 (0.25) | ~~95.47~~ (1.04) | 90.41 (0.16) | |
| | **0.34** (0.00) | 0.39 (0.01) | 0.65 (0.03) | 2.31 (0.03) | **0.34** (0.00) | |
| protein | 90.07 (0.10) | 92.01 (0.10) | 91.46 (0.31) | ~~97.94~~ (0.90) | 91.87 (0.18) | |
| | **1.59** (0.00) | 1.64 (0.00) | 1.86 (0.03) | 3.81 (0.03) | 1.61 (0.01) | |

the empirical coverage when the test set is divided into subgroups based on interval width. In the left of the two histograms, we observe that the baseline method, OQR, under-covers for narrower intervals whilst over-covering for wider intervals. This is in contrast to QFR-O on the right which achieves generally balanced coverage across all interval widths. Given that conditional coverage is the exclusive goal in this case, the QFR-O solution discovers that wider intervals are necessary to achieve this. As a result, it is able to ensure that the probability of error is not dependent on interval width resulting in a more consistent solution.

Therefore, unlike previous works, QRF may be viewed as a general-purpose approach to constructing intervals in which the practitioner may choose to prioritize among additional interval properties depending on a particular application. In this robotics example, minimizing interval width may result in a more accurate estimate of the object's distance while improving conditional coverage may help prevent specific failure modes.

**Benchmarking QFR-W.** We now proceed to investigate the performance of QFR-W on the standard quantile regression benchmark tasks used in Tagasovska & Lopez-Paz (2019); Chung et al. (2021); Pearce et al. (2018) consisting of nine datasets from Asuncion & Newman (2007). We follow the preprocessing and experimental protocol described in Appendix C in line with previous works. To summarize, we train two-layer neural networks using a grid search to find optimal hyperparameters. All experiments are repeated over 20 seeds with means and standard errors of means reported throughout. For building SQR intervals, we followed the method prescribed by the authors which consists of selecting the symmetric (0.05, 0.95) intervals. The results are included in Table 3 where we provide the resulting coverage and MPIW for each dataset. We also include histograms of the target distributions for each dataset to highlight the point that non-symmetric distributions should be the *standard expectation* on real-world regression tasks. Of course, shorter intervals are only desirable when the target level of coverage is maintained. Therefore, we exclude results that fail to achieve coverage which we indicate with a ~~strikethrough~~. In line with previous work of Tagasovska & Lopez-Paz (2019), we consider coverage to be met if the empirical coverage lies within 2.5% of the target level $\alpha$ (after accounting for uncertainty). While obtaining empirical coverage that is greater than the desired coverage level may often be less harmful than obtaining less than the desired coverage level, at a minimum this still reflects an inefficiency. Additionally, there are many applications in which we are primarily interested in the miscoverage cases (e.g. extreme events)

Table 4: **Benchmarking QFR-O.** We compare our proposed loss function to those used in OQR for achieving improved conditional coverage as evaluated using standard metrics (see Section 2). We report the empirical coverage achieved and the % improvement obtained over OQR in Pearson correlation and HSIC ($\pm$ a standard error).

| Dataset | QFR-O (ours) coverage | OQR coverage | Pearson correlation | HSIC |
|---------|----------------------|--------------|---------------------|------|
| concrete | 88.75 ($\pm$0.14) | 87.54 ($\pm$0.12) | +80.19 ($\pm$0.72) | +98.01 ($\pm$0.12) |
| power | 90.00 ($\pm$0.05) | 91.86 ($\pm$0.05) | +86.49 ($\pm$0.63) | +77.79 ($\pm$1.19) |
| wine | 89.15 ($\pm$0.27) | 88.59 ($\pm$0.10) | +71.42 ($\pm$1.14) | +96.77 ($\pm$0.29) |
| yacht | 88.71 ($\pm$0.99) | 89.07 ($\pm$0.20) | -45.80 ($\pm$11.28) | -18.49 ($\pm$10.7) |
| naval | 90.58 ($\pm$0.07) | 90.12 ($\pm$0.07) | +87.83 ($\pm$0.48) | +16.87 ($\pm$2.95) |
| energy | 89.77 ($\pm$0.18) | 90.42 ($\pm$0.09) | +25.04 ($\pm$2.70) | +76.57 ($\pm$2.12) |
| boston | 90.49 ($\pm$0.18) | 91.97 ($\pm$0.15) | +75.97 ($\pm$0.80) | +85.67 ($\pm$1.54) |
| kin8nm | 90.40 ($\pm$0.07) | 89.94 ($\pm$0.05) | +89.16 ($\pm$0.30) | +99.91 ($\pm$0.01) |
| protein | 89.87 ($\pm$0.02) | 89.45 ($\pm$0.03) | +80.29 ($\pm$0.58) | +99.91 ($\pm$0.01) |

and, therefore, overcoverage may be problematic in addition to being inefficient. Thus, this protocol symmetrically discards intervals that undercover and overcover by a certain margin. The target coverage level is set to 90% throughout our experiments. These results demonstrate that the added flexibility of circumventing the standard step of learning predefined quantiles can be utilized to obtain narrower intervals in practice.

**Benchmarking QFR-O.** In a similar vein, we evaluate QFR-O for its effectiveness in improving measures of conditional coverage. In this case, we compare to the original OQR work of Feldman et al. (2021). We evaluate both methods on the same benchmark datasets as previously and follow the same experimental protocol as in Feldman et al. (2021). The key distinction in this experiment is that the coefficient of the regularization parameter for both methods is incrementally reduced until empirical coverage is achieved. This is because a comparison of conditional coverage using these metrics is only meaningful if both methods achieve a similar level of empirical marginal coverage. Again, the experimental setup is described in detail in Appendix C. We investigate this regularization term's effectiveness at achieving its stated objective of enforcing orthogonality between interval width and instances of miscoverage. Since both methods use an identical regularization expression, the gains obtained by QFR-O in this section are due to pairing this term with our QFR objective from Equation (1) rather than existing quantile regression objectives that suffer from the limitations discussed in Section 3.1. The results are provided in Table 4 where we compare performance based on a test set evaluation of coverage, % improvement in Pearson correlation over OQR, and % improvement in HSIC over OQR. We generally find that QFR-O achieves a significant improvement in these measures of conditional coverage, again indicating that the added flexibility of this approach enables a more favorable solution to be found. An important takeaway from these results is that, since Pearson's correlation is the regularization objective used by both methods, the gains in performance when considering it as an evaluation metric provide direct evidence that the added flexibility provided by the OQR loss function (due to not being centered around the median) enables it to find a better solution in the auxiliary task (in this case, minimizing Pearson's correlation between instances of miscoverage and interval width).

## 5 CONCLUSION

In this work we have introduced *Quantile-Free Regression*, a direct alternative to quantile regression that circumvents the requirement to prespecify the specific quantiles being learned whilst maintaining its attractive coverage properties. We then demonstrated that this new loss can be easily combined with user-specified regularization terms to obtain a solution suitable for a given application (i.e. narrower intervals or improved conditional coverage). Finally, we evaluated the method against state-of-the-art single-model methods across standard benchmark tasks demonstrating that this added flexibility is converted into improved performance in terms of either narrower intervals or better conditional coverage, depending on the practitioner's preference. For future work, devising new approaches to evaluating predictive intervals and developing complimentary regularization terms such that they can be paired with QFR provides a promising direction.

REPRODUCABILITY STATEMENT

We have endeavored to ensure that the experiments in this work are reproducible. A detailed description of the experiments performed in Section 4 is provided in Appendix C. We have also provided the code implementation in an attached zip file which we intend to also host on GitHub upon acceptance.

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

# A    EXTENDED RELATED WORK

In this work, we have focused on *single model approaches* (i.e. quantile/interval regression) for obtaining predictive intervals from neural networks with the intention of developing more effective methods within this category. Due to the vital importance of uncertainty quantification, a vast literature of disparate alternative approaches has emerged, each representing interesting research directions with their own respective strengths and weaknesses. In this section, we provide a broad overview of the leading methods for obtaining predictive intervals from neural networks. Given the extensive nature of this topic, this summary is not exhaustive and we refer the reader to the more comprehensive references cited within each topic for a more complete overview. For a recent survey on predictive intervals in regression problems more generally (i.e. beyond neural networks) we refer the reader to Dewolf et al. (2023) or for general neural network uncertainty quantification to Gawlikowski et al. (2023).

**Single model approaches.** The subject of this work is individual models that output intervals rather than point estimates which we refer to as single model approaches. An in depth description of the state of the art methods within this category is provided in Section 2. Whilst these methods typically estimate aleatoric uncertainty, it is quite straightforward to combine them with Bayesian or ensemble methods to simultaneously account for epistemic uncertainty (Tagasovska & Lopez-Paz, 2019). As previously discussed, quantile regression (Koenker & Bassett Jr, 1978) has excelled in this category – including prior to and outside of the deep learning regime (Koenker & Hallock, 2001; Meinshausen & Ridgeway, 2006; Yu & Jones, 1998). Similarly, the evaluated direct interval estimation method of Pearce et al. (2018) was also an evolution of the foundational work of Khosravi et al. (2010). Elsewhere in the time series setting, Gasthaus et al. (2019) estimate quantiles using monotonic regression splines. Whilst strictly speaking a dual model approach, variance networks fit a second neural network to estimate the variance of a prediction (Skafte et al., 2019). A very simple approach that can be easily combined with most other approaches is to regularize the neural network such that measures of calibration are optimized using e.g. confident output penalization (Pereyra et al., 2017), label smoothing (Szegedy et al., 2016), or induced label noise (Xie et al., 2016).

**Bayesian methods.** Bayesian neural networks attempt to model the target probability distribution for a given test example $x$ given some observed data $\mathcal{D}$ by marginalizing over a distribution of network parameters $\theta$ such that $P(y|x, \mathcal{D}) = \int P(y|x, \theta)P(\theta|\mathcal{D})$ from which predictive intervals can be derived. This expression requires intractable calculations which may be approximated in practice using various techniques. A simple approach consists of taking a second-order Taylor expansion around the maximum a posteriori estimate of $\theta$ to produce a Gaussian approximation of $P(\theta|\mathcal{D})$ known as the *Laplace approximation* (Tierney & Kadane, 1986). In practice, further scaling efforts have been required to apply this method in the modern deep learning context (Daxberger et al., 2021). Alternatively, *variational inference* substitutes $P(\theta|\mathcal{D}) \approx q(\theta)$ where $q(\theta)$ denotes some tractable parametric approximation such as a Gaussian distribution (Hinton & Van Camp, 1993). Significant research has investigated methods for extending this approach to account for modern datasets and architectures (Graves, 2011; Zhang et al., 2018). *Monte Carlo integration* instead approximates the integral over parameters with a finite sum such that $\int P(y|x, \theta)P(\theta|\mathcal{D}) \approx \frac{1}{M} \sum_{j=1}^{M} P(y|x, \theta_j)$ (Caflisch, 1998). Then different choices of selecting a subset of $M$ weight parameterizations result in alternative instantiations of this approximation (see e.g. Ch. 17 of Goodfellow et al. (2016)). *Monte Carlo dropout*, which drops neurons at test time according to a Bernoulli distribution to estimate uncertainty, has become popular due to its conceptual and implementation simplicity (Gal & Ghahramani, 2016). A somewhat distinct Bayesian approach is the *Gaussian process* which is a collection of random variables of which any finite sample is Gaussian distributed specified by a specific mean function and covariance kernel (Rasmussen et al., 2006). Although this method suffers from important limitations (e.g. scaling), its adaption to the deep learning setting has achieved notable performance on benchmark tasks (Wilson et al., 2016b;a).

**Deep ensembles.** Aggregating outputs over a set of neural networks has emerged as a simple but effective method that accounts for epistemic uncertainty (Lakshminarayanan et al., 2017). Whilst this approach can be considered studied through a Bayesian perspective (Wilson & Izmailov, 2020; Wilson, 2020), it has primarily developed from the classical ensembling literature (Sagi & Rokach, 2018). It is hypothesised that the empirical success of deep ensembles is due to their better exploration of the loss landscape (Fort et al., 2019) – with diversity typically achieved through random

Table 5: **Categories of interval construction approaches.** A broad comparison of different categories of interval construction highlighting the key distinctions of single model approaches. Detailed descriptions and discussion is provided in the text.

|  | (1) | (2) | (3) | (4) |
|---|---|---|---|---|
| Single model approaches | ✓ | ✓ | ✓ | ✓ |
| Bayesian methods | ✗ | ✗ | ✓ | ½ |
| Deep ensembles | ✗ | ✗ | ✗ | ✓ |
| Parametric approaches | ½ | ✗ | ✓ | ½ |
| Post-hoc methods | ✓ | ✓ | ✗ | ✓ |

initialization and batching due to known challenges in optimizing for exploration (Jeffares et al., 2023; Abe et al., 2023). However, recent works have suggested that their improved calibration may be overstated (Rahaman et al., 2021) and that performance increases may be better understood as being due to an increased model capacity (Abe et al., 2022). These gains also come at the cost of a significant computational overhead with the relative computational cost growing linearly with ensemble size. More efficient approaches to deep ensembling have been proposed in recent years to reduce this cost (e.g. Wen et al., 2019).

**Parametric approaches.** The classical statistics literature has a long history of developing principled estimates of predictive intervals derived from parametric assumptions in the data generating process (see e.g. Ch. 5.3 Seber & Lee, 2003). One such approach in the neural network setting is the delta method which makes a linearity assumption in the region around a prediction paired with a Gaussian assumption on the noise distribution (Hwang & Ding, 1997; Khosravi et al., 2011). Another example is Nix & Weigend (1994) who also assume the noise distribution to be Gaussian and derive a cost function to estimate its value with an auxiliary output to the network.

**Post-hoc methods.** Several methods exist in which calibrated intervals are constructed or updated as a post-processing step for an existing point predictor or interval estimator respectively. Perhaps the most notable of these is conformal prediction (Vovk et al., 2005) and, in particular, inductive conformal prediction (Papadopoulos, 2008), which provides prediction intervals with finite sample marginal coverage guarantees. Romano et al. (2019) further developed an approach that also performs well conditionally (i.e. intervals where width is adaptive to a given example). As noted by the authors, this method "can wrap around any algorithm for quantile regression", thus making it a complimentary post-hoc approach. Other methods typically focus on improving the calibration of an underlying point predictor. Approaches include Platt scaling (Platt et al., 1999), temperature scaling (Tomani et al., 2021), histogram binning (Zadrozny & Elkan, 2001), test time augmentation (Hekler et al., 2023), and isotonic regression (Zadrozny & Elkan, 2002).

**Comparing categories of interval construction.** We now provide some high level distinctions between these categories of approaches for constructing intervals. This is *not* intended as a complete evaluation of competing uncertainty quantification approaches, rather we wish to highlight the advantages of single model approaches to emphasize the significance of developments *within* this category. Furthermore, due to the broadness of these categories and the lack of clear boundaries between them, the following distinctions act as generalizations for which some exceptions exist. A discussion of these distinctions is provided in the next paragraph with a summary provided in Table 5.

**(1)** *Minimal computational overhead* - Deep ensembles require training a neural network from scratch $M$ times while Bayesian methods typically require approximations to produce tractable algorithms for large-scale models (Abdullah et al., 2022; Osawa et al., 2019). Parametric methods require some overhead with the specific amount method dependent. In contrast, the other categories, including single model approaches, generally only require at most a change of loss function at training time. **(2)** *Data-assumption-free valid intervals* - Parametric approaches and Bayesian methods

generally require assumptions on the data-generating process to provide validity guarantees on their intervals. Deep ensembles don't provide such guarantees on derived intervals. However, many single model and post-hoc methods provide asymptotic or even finite sample guarantees of valid intervals. **(3)** *Distinguishes between aleatoric & epistemic uncertainty* - Explicitly differentiating between aleatoric (irreducible) and epistemic (reducible) uncertainty can be valuable (Hüllermeier & Waegeman, 2021). As discussed in Tagasovska & Lopez-Paz (2019); Pearce et al. (2018), the loss function of single model approaches generally estimates aleatoric uncertainty while epistemic uncertainty can also be accounted for by applying e.g. orthonormal certificates or interval ensembling. Explicitly modeling uncertainties in this way also tends to be at the heart of Bayesian and parametric methods (see e.g. Kendall & Gal (2017)). Deep ensembles and post-hoc methods do not typically make this distinction. **(4)** *Directly applicable to any loss-based algorithm* - Single-model approaches are typically characterized by simply replacing a point estimate loss function with a quantile regression loss (e.g. mean squared error → pinball loss). Similarly, deep ensembles only require running an algorithm multiple times and post-hoc methods typically wrap around or recalibrate arbitrary models. Some Bayesian and parametric methods are generally applicable (e.g. Monte Carlo dropout) however others require more substantial changes resulting in different algorithms when applied to neural networks (e.g. a Gaussian process).

# B PROOFS

**Theorem 3.1** (QFR Coverage). *For any random variable $Y$ associated with an input $x$, $\forall\, \alpha \in [0, 1]$,*

$$(\mu_1^*(x), \mu_2^*(x)) = \underset{\mu_1,\mu_2}{\arg\min} \{\mathbb{E}_Y(\mathcal{L}_\alpha^{QFR}((\mu_1, \mu_2), x, Y))\} \implies \mathbb{P}(\mu_1^*(x) < Y < \mu_2^*(x)) = \alpha$$

*Proof.* In these proofs, we omit to precise the input $x$ every line for clarification purposes. However, the reader should not forget that Y is the random variable associated with the input x.

First, we can rewrite our new loss with an indicator function :

$$\mathcal{L}_\alpha^{\text{QFR}}((\mu_1, \mu_2), y)) = (y - \mu_1)(y - \mu_2)(\alpha - \mathbb{I}_{y \in [\mu_1, \mu_2]})$$

Then, we consider the expectation of the loss :

$$\mathbb{E}(\mathcal{L}_\alpha^{\text{QFR}}((\mu_1, \mu_2), Y))) = \alpha \int_{-\infty}^{\infty} (y - \mu_1)(y - \mu_2) d\mathbb{P}_Y(y) - \int_{\mu_1}^{\mu_2} (y - \mu_1)(y - \mu_2) d\mathbb{P}_Y(y)$$

We find the expression of its partial derivatives with respect to the bounds:

$$\frac{\partial \mathbb{E}(\mathcal{L}_\alpha^{\text{QFR}})}{\partial \mu_1} = -\alpha \int_{-\infty}^{\infty} (y - \mu_2) d\mathbb{P}_Y(y) + \int_{\mu_1}^{\mu_2} (y - \mu_2) d\mathbb{P}_Y(y)$$

$$\frac{\partial \mathbb{E}(\mathcal{L}_\alpha^{\text{QFR}})}{\partial \mu_2} = -\alpha \int_{-\infty}^{\infty} (y - \mu_1) d\mathbb{P}_Y(y) + \int_{\mu_1}^{\mu_2} (y - \mu_1) d\mathbb{P}_Y(y)$$

At the minimum, the gradient of the expected loss is null. Thus, if there is a minimum at the point $(\mu_1^*, \mu_2^*)$ with $\mu_2^* > \mu_1^*$,

$$\frac{\partial \mathbb{E}(\mathcal{L}_\alpha^{\text{QFR}})}{\partial \mu_1}\bigg|_{\mu_1^*,\mu_2^*} - \frac{\partial \mathbb{E}(\mathcal{L}_\alpha^{\text{QFR}})}{\partial \mu_2}\bigg|_{\mu_1^*,\mu_2^*} = 0$$

$$\implies -\alpha \int_{-\infty}^{\infty} (\mu_1^* - \mu_2^*) d\mathbb{P}_Y(y) + \int_{\mu_1^*}^{\mu_2^*} (\mu_1^* - \mu_2^*) d\mathbb{P}_Y(y) = 0$$

$$\implies -\alpha(\mu_1^* - \mu_2^*) \int_{-\infty}^{\infty} d\mathbb{P}_Y(y) + (\mu_1^* - \mu_2^*) \int_{\mu_1^*}^{\mu_2^*} d\mathbb{P}_Y(y) = 0$$

$$\implies -\alpha \int_{-\infty}^{\infty} d\mathbb{P}_Y(y) + \int_{\mu_1^*}^{\mu_2^*} d\mathbb{P}_Y(y) = 0$$

$$\implies \mathcal{P}(\mu_1^* < Y < \mu_2^*) = \alpha$$

$\square$

**Theorem 3.2** (QFR with finite samples). *For any random variable $Y$ associated with an input $x$, we consider $N$ realizations of this random variable : $\{y_i\}_{i=1,N}$. $\forall \alpha \in [0,1]$ such that $\alpha \cdot N \in \mathbb{N}$,*

$$(\mu_1^*(x), \mu_2^*(x)) = \underset{\mu_1, \mu_2}{\arg\min} \{\sum_{i=1}^{N} \mathcal{L}_\alpha^{QFR}((\mu_1, \mu_2), x, y_i)\} \implies \frac{1}{N}\sum_{i=1}^{N} \mathbb{I}_{y_i \in [\mu_1*, \mu_2*]} = \alpha$$

*Proof.* We rewrite our sum as an integral with discrete density $\mathbb{I}_{y \in \{y_i\}} dy$ :

$$\sum_{i=1}^{N} \mathcal{L}_\alpha^{\text{QFR}}((\mu_1, \mu_2), x, y_i) = \alpha \int_{-\infty}^{\infty} (y - \mu_1)(y - \mu_2)\mathbb{I}_{y \in \{y_i\}}dy - \int_{\mu_1}^{\mu_2} (y - \mu_1)(y - \mu_2)\mathbb{I}_{y \in \{y_i\}}dy$$

We find the expression of its partial derivatives with respect to the bounds:

$$\frac{\partial \sum_{i=1}^{N} \mathcal{L}_\alpha^{\text{QFR}}((\mu_1, \mu_2), x, y_i)}{\partial \mu_1} = -\alpha \int_{-\infty}^{\infty} (y - \mu_2)\mathbb{I}_{y \in \{y_i\}}dy + \int_{\mu_1}^{\mu_2} (y - \mu_2)\mathbb{I}_{y \in \{y_i\}}dy$$

$$\frac{\partial \sum_{i=1}^{N} \mathcal{L}_\alpha^{\text{QFR}}((\mu_1, \mu_2), x, y_i)}{\partial \mu_2} = -\alpha \int_{-\infty}^{\infty} (y - \mu_1)\mathbb{I}_{y \in \{y_i\}}dy + \int_{\mu_1}^{\mu_2} (y - \mu_1)\mathbb{I}_{y \in \{y_i\}}dy$$

At the minimum, the gradient of the expected loss is null. Thus, if there is a minimum at the point $(\mu_1^*, \mu_2^*)$ with $\mu_2^* > \mu_1^*$,

$$\frac{\partial \sum_{i=1}^{N} \mathcal{L}_\alpha^{\text{QFR}}((\mu_1, \mu_2), x, y_i)}{\partial \mu_1}\Big|_{\mu_1^*, \mu_2^*} - \frac{\partial \sum_{i=1}^{N} \mathcal{L}_\alpha^{\text{QFR}}((\mu_1, \mu_2), x, y_i)}{\partial \mu_2}\Big|_{\mu_1^*, \mu_2^*} = 0$$

$$\implies -\alpha \int_{-\infty}^{\infty} (\mu_1^* - \mu_2^*)\mathbb{I}_{y \in \{y_i\}}dy + \int_{\mu_1^*}^{\mu_2^*} (\mu_1^* - \mu_2^*)\mathbb{I}_{y \in \{y_i\}}dy = 0$$

$$\implies -\alpha(\mu_1^* - \mu_2^*)\sum_{i=1}^{N} 1 + (\mu_1^* - \mu_2^*)\sum_{y_i \in [\mu_1^*, \mu_2^*]} 1 = 0$$

$$\implies \frac{1}{N}\sum_{i=1}^{N} \mathbb{I}_{y_i \in [\mu_1*, \mu_2*]} = \alpha$$

$\square$

**Theorem 3.3** (QFR-W Coverage). *For any random variable $Y$ associated with an input $x$, $\forall \alpha \in [0,1]$,*

$$(\mu_1^*(x), \mu_2^*(x)) = \underset{\mu_1, \mu_2}{\arg\min} \{\mathbb{E}_Y(\mathcal{L}_{\alpha+2\lambda}^{QFR-W}(\mu_1, \mu_2), x, Y))\} \implies \mathbb{P}(\mu_1^*(x) < Y < \mu_2^*(x)) = \alpha$$

*Proof.* Similarly, we are starting by rewriting the QFR-W loss with indicator functions.

$$\text{QFR-W}_\alpha(\mu_1, \mu_2, y) = \text{QFR}_\alpha(\mu_1, \mu_2, y) + \frac{\lambda(\mu_2 - \mu_1)^2}{2}$$

$$= (y - \mu_1)(y - \mu_2)(\alpha - \mathbb{I}_{y \in [\mu_1, \mu_2]}) + \frac{\lambda(\mu_2 - \mu_1)^2}{2}$$

Then, we consider the expectation of the loss :

$$\mathbb{E}(\text{QFR-W}_\alpha(\mu_1, \mu_2, Y)) =$$
$$\alpha \int_{-\infty}^{\infty} (y - \mu_1)(y - \mu_2)d\mathbb{P}_Y(y) - \int_{\mu_1}^{\mu_2} (y - \mu_1)(y - \mu_2)d\mathbb{P}_Y(y) + \frac{\lambda(\mu_2 - \mu_1)^2}{2}$$

We find the expression of the partial derivatives with respect to the bounds:

$$\frac{\partial \mathbb{E}(\text{QFR-W}_\alpha)}{\partial \mu_1} = -\alpha \int_{-\infty}^{\infty} (y - \mu_2) d\mathbb{P}_Y(y) + \int_{\mu_1}^{\mu_2} (y - \mu_2) d\mathbb{P}_Y(y) - \lambda(\mu_2 - \mu_1)$$

$$\frac{\partial \mathbb{E}(\text{QFR-W}_\alpha)}{\partial \mu_2} = -\alpha \int_{-\infty}^{\infty} (y - \mu_1) d\mathbb{P}_Y(y) + \int_{\mu_1}^{\mu_2} (y - \mu_1) d\mathbb{P}_Y(y) + \lambda(\mu_2 - \mu_1)$$

At the minimum, the gradient of the expected loss is null. Thus, if there is a minimum at the point $(\mu_1^*, \mu_2^*)$ with $\mu_2^* > \mu_1^*$,

$$\frac{\partial \mathbb{E}(\text{QFR-W}_\alpha)}{\partial \mu_1}\bigg|_{\mu_1^*, \mu_2^*} - \frac{\partial \mathbb{E}(\text{QFR-W}_\alpha)}{\partial \mu_2}\bigg|_{\mu_1^*, \mu_2^*} = 0$$

$$\implies -\alpha \int_{-\infty}^{\infty} (\mu_1^* - \mu_2^*) d\mathbb{P}_Y(y) + \int_{\mu_1^*}^{\mu_2^*} (\mu_1^* - \mu_2^*) d\mathbb{P}_Y(y) + 2\lambda(\mu_1^* - \mu_2^*) = 0$$

$$\implies -\alpha(\mu_1^* - \mu_2^*) \int_{-\infty}^{\infty} d\mathbb{P}_Y(y) + (\mu_1^* - \mu_2^*) \int_{\mu_1^*}^{\mu_2^*} d\mathbb{P}_Y(y) + 2\lambda(\mu_1^* - \mu_2^*) = 0$$

$$\implies -\alpha \int_{-\infty}^{\infty} d\mathbb{P}_Y(y) + \int_{\mu_1^*}^{\mu_2^*} d\mathbb{P}_Y(y) + 2\lambda = 0$$

$$\implies \mathcal{P}(\mu_1^* < Y < \mu_2^*) = \alpha - 2\lambda$$

Then, as $\alpha$ is a constant, we can replace it with a corrected term. When choosing $\hat{\alpha} = \alpha + 2\lambda$, we obtain

$$(\mu_1^*, \mu_2^*) = \underset{\mu_1, \mu_2}{\arg\min} \{\mathbb{E}(\text{QFR-W}_{\hat{\alpha}}(\mu_1, \mu_2, Y))\} \implies \mathcal{P}(\mu_1^* < Y < \mu_2^*) = \hat{\alpha} - 2\lambda = \alpha$$

$\square$

**Proposition 3.1** (Existence and Uniqueness of Solution). *$\mu_1^{min}$ and $\mu_2^{max}$ denote the boundaries of our optimization problem. For a target distribution $Y$ with a cumulative distribution function that is $k$-Lipschitz continuous with $k < 1 + \frac{\alpha}{\mu_2^{max} - \mu_1^{min}}$, when $\lambda > max(0, \int_{\mu_1^{min}}^{\mu_2^{max}} d\mathbb{P}_Y(y) - \alpha)$, the minimum of $\mathcal{L}_{\alpha+2\lambda}^{QFR-W}$ exists and is unique.*

*Proof.* We study the function $\mu_1, \mu_2 \mapsto \mathbb{E}(\mathcal{L}_\alpha^{\text{QFR-W}}((\mu_1, \mu_2), Y))$ in a closed subset of $\mathbb{R}^2$ where $\mu_1 < \mu_2$. We named this subset $\mathbb{D}$. On this closed subset of the space to say, it exists $\mu_2^{max}$ and $\mu_1^{min}$ such that $\forall (\mu_1, \mu_2) \in \mathbb{D}$ $\mu_2 < \mu_2^{max}$ and $\mu_1 > \mu_1^{min}$.

Moreover, we assume that Y can be associated with a probability density function $d\mathbb{P}_Y$ and that its cumulative distribution function is k-Lipschitz continuous with $k < 1 + \frac{\alpha}{\mu_2^{max} - \mu_1^{min}}$

$$\forall (\mu_2, \mu_1) \in [\mu_1^{min}, \mu_2^{max}]^2 \mid \int_{\mu_1}^{\mu_2} d\mathbb{P}_Y(y)| < (1 + \frac{\alpha}{\mu_2^{max} - \mu_1^{min}})|(\mu_2 - \mu_1)|$$

The non-negativity of the studied function gives us the existence of the minimum.

To demonstrate the uniqueness of the minimum, we show that the eigenvalues of the Hessian matrix are positive. We start by computing the gradient of the expected loss :

$$\nabla \mathbb{E}(\mathcal{L}_\alpha^{\text{QFR-W}}) = \begin{bmatrix} \frac{\partial \mathbb{E}(\mathcal{L}_\alpha^{\text{QFR-W}})}{\partial \mu_1} \\ \frac{\partial \mathbb{E}(\mathcal{L}_\alpha^{\text{QFR-W}})}{\partial \mu_2} \end{bmatrix}$$

$$= \begin{bmatrix} -\alpha \int_{-\infty}^{\infty} (y - \mu_2) d\mathbb{P}_Y(y) + \int_{\mu_1}^{\mu_2} (y - \mu_2) d\mathbb{P}_Y(y) - \lambda(\mu_2 - \mu_1) \\ -\alpha \int_{-\infty}^{\infty} (y - \mu_1) d\mathbb{P}_Y(y) + \int_{\mu_1}^{\mu_2} (y - \mu_1) d\mathbb{P}_Y(y) + \lambda(\mu_2 - \mu_1) \end{bmatrix}$$

Then, we compute the Hessian matrix :

$$\nabla^2 \mathbb{E}(\mathcal{L}_\alpha^{\text{QFR-W}}) = \begin{bmatrix} \mu_2 - \mu_1 + \lambda & \alpha - \int_{\mu_1}^{\mu_2} d\mathbb{P}_Y(y) - \lambda \\ \alpha - \int_{\mu_1}^{\mu_2} d\mathbb{P}_Y(y) - \lambda & \mu_2 - \mu_1 + \lambda \end{bmatrix}$$

The eigenvalues of the Hessian matrix are given by $\lambda_\pm = \mu_2 - \mu_1 + \lambda \pm |\alpha - \int_{\mu_1}^{\mu_2} d\mathbb{P}_Y(y) - \lambda|$

From that, we get that $\lambda > \int_{\mu_1^{min}}^{\mu_2^{max}} d\mathbb{P}_Y(y) - \alpha \implies \forall(\mu_1, \mu_2) \in \mathbb{D}\ \lambda > \int_{\mu_1}^{\mu_2} d\mathbb{P}_Y(y) - \alpha$ (1) because the probability density function $d\mathbb{P}_Y$ is positive.

Additionally, we use the assumption of the k-Lipschitz continuity of the CDF and we obtain the following inequality :

$$\forall(\mu_1, \mu_2) \in \mathbb{D}\ \mu_2 - \mu_1 \geq \int_{\mu_1}^{\mu_2}(1 + \frac{\alpha}{\mu_2 - \mu_1}) - \alpha \geq \int_{\mu_1}^{\mu_2}(1 + \frac{\alpha}{\mu_2^{max} - \mu_1^{min}}) - \alpha \geq \int_{\mu_1}^{\mu_2} d\mathbb{P}_Y - \alpha$$

Therefore, under the condition $\lambda > \max(0, \int_{\mu_1^{min}}^{\mu_2^{max}} d\mathbb{P}_Y(y) - \alpha)$, we obtain the positiveness of $\lambda_-$ :

$$\lambda_- = \mu_2 - \mu_1 + \lambda - |\alpha - \int_{\mu_1}^{\mu_2} d\mathbb{P}_Y(y) - \lambda| = \mu_2 - \mu_1 + \lambda - (\lambda - \alpha + \int_{\mu_1}^{\mu_2} d\mathbb{P}_Y(y))\ (1)$$

$$\implies \lambda_- = \mu_2 - \mu_1 - (-\alpha + \int_{\mu_1}^{\mu_2} d\mathbb{P}_Y(y)) > 0$$

The first eigenvalue $\lambda_+$ is obviously non-negative, thus both eigenvalues are non-negative which means that the Hessian matrix is semi-definite positive.

In conclusion, when the condition $\lambda > \max(0, \int_{\mu_1^{min}}^{\mu_2^{max}} d\mathbb{P}_Y(y) - \alpha)$ is respected, our optimal interval prediction loss is convex. Hence, it has a unique minimum.

Both the existence and the uniqueness of the minimum have been proven. $\qquad\square$

**Theorem 3.4** (Validity of QFR-O - A variation of the validity of orthogonal quantile regression theorem from (Feldman et al., 2021)). *Suppose $Y|X = x$ follows a continuous distribution for each $x \in \mathcal{X}$, and suppose that $\mu_1(X), \mu_2(X) \in \mathcal{F}$. Consider the infinite-data version of the QFR-O optimization :*

$$\underset{\mu_1, \mu_2 \in \mathbb{F}}{\arg\min} \{\mathbb{E}(\mathcal{L}_\alpha^{QFR}((\mu_1, \mu_2), X, Y) + \gamma \mathbb{R}(w, m))\}$$

*Then, true conditional intervals with $\alpha$ coverage are solutions to the above optimization problem.*

*Proof.* We note $m = \mathbb{I}_{y \in [\mu_1, \mu_2]}$ the coverage function and $w = |\mu_2 - \mu_1|$ the interval length function. We consider the true conditional intervals that satisfied $\mathbb{P}(\mu_1(X) < Y < \mu_2(X)|X = x) = \alpha$. Feldman et al. (2021) has shown that for such intervals, the coverage and interval length functions are independent.

Therefore, similarly to vanilla QR or the Winkler score, these true conditional intervals are a solution to the QFR problem. Moreover, by definition, the independence of $w$ and $m$ fixes the Pearson correlation (or the HSIC score) of these functions to 0. Thus, the orthogonal penalty term that is based on these metrics is also minimized. Hence, the true conditional intervals with $\alpha$ coverage are a solution to the QFR-0 optimization problem. $\qquad\square$

## C EXPERIMENTAL DETAILS

We follow standard preprocessing on all datasets with features and targets standardized such that they have zero mean and unit variance. All experiments are repeated over 20 random seeds with means and standard errors of means reported throughout. We train two-layer neural networks with

64 hidden units and ReLU activations throughout (consistent with (Feldman et al., 2021)). We also use the Adam optimizer (Kingma & Ba, 2014).

In the benchmarking of QFR-W, we applied the following experimental design. A training-validation-testing split with a ratio of 0.2 for testing and a further split of 0.95-0.05 for training-validation is applied. Then we perform a hyperparameter grid search for all methods. Each combination is first fit to the training data with the evaluation performed on the validation data. For a given hyperparameter combination, the best model is selected based on the epoch that achieves the best interval length (such that target coverage is achieved). Once the model has converged, it is retrained on the combined training and validation data for the same number of epochs with the final reported evaluation on the held-out test set. All methods evaluated follow this protocol with the exception of SQR which we found to be unstable resulting in poor performance due to the smaller datasets and its aforementioned more difficult learning problem. Therefore, SQR uses just a single training-testing split. The grid search considers the following hyperparameters: dropout probability $\in \{0, 0.1, 0.2, 0.3, 0.4\}$, maximum number of epochs $\in \{1000, 2000, 3000, 5000\}$, batch size $\in \{64, 512, 1014, 2048\}$, exponential learning rate schedule $\in \{0.995, 1\}$, learning rate $\in [5e-1, 5e-5]$, and regularization coefficient $\in [5e-4, 5e+2]$. The learning rate started at 0.0005 and incrementally increased and decreased until the bound was reached or the learning dynamics became unstable and failed to converge. For the regularization coefficient, the same procedure was applied, starting at 0.1 and incrementally increasing and decreasing until the solution has stopped achieving the desired coverage on the validation set or the bound is reached.

In the benchmarking of QFR-O, we followed the exact procedure and used the implementation of the OQR baseline in Feldman et al. (2021). In what follows, we will describe this procedure. A training-validation-testing split with a ratio of 0.4 for testing and a further split of 0.9-0.1 for training-validation is applied. In this case, the model is not retrained and is simply evaluated on the test set after converging on the validation set. The default hyperparameters are used for both methods with only the regularization coefficient tuned. Specifically, it is set to 1 and then decreased in increments following $[1, 0.5, 0.1, 0.05, \ldots]$ until the desired coverage is achieved. Both the QFR-O and OQR hyperparameters are - learning rate: 1e-3, maximum number of epochs: 10000, dropout probability: 0, and batch size: 1024. Early stopping patience is set to 200 epochs.

The code used to run these experiments is provided in the attached zip file and will be uploaded to GitHub upon acceptance.

Table 6: Summary statistics of the standard benchmark datasets for quantile regression.

| Dataset | Mean | Variance | Skewness | Kurtosis |
|---------|------|----------|----------|----------|
| concrete | 35.82 | 279.08 | 0.42 | -0.32 |
| wine | 5.64 | 0.65 | 0.22 | 0.29 |
| yacht | 10.50 | 229.84 | 1.75 | 2.00 |
| energy | 22.31 | 101.81 | 0.36 | -1.25 |
| kin8nm | 0.71 | 0.07 | 0.09 | -0.53 |
| naval | 0.99 | 0.00 | -0.00 | -1.20 |
| power | 454.37 | 291.28 | 0.31 | -1.05 |
| boston | 22.53 | 84.59 | 1.10 | 1.47 |
| protein | 7.75 | 37.43 | 0.57 | -1.14 |

## D    FORMAL DESCRIPTION OF EVALUATION METRICS

To ensure this work is self-contained, in this section we include a formal description of the evaluation metrics introduced in Section 2 and used in Section 4. We implement these metrics as described in the referenced works. In all cases we evaluate on some evaluation dataset $\mathcal{D}$ consisting of $N$ input/target pairs $(\mathbf{x}, y)$.

**Definition 1** (Prediction Interval Coverage Probability (PICP) (Tagasovska & Lopez-Paz, 2019))**.**
*Defined as the number of true observations falling inside the estimated prediction interval, this is*

*calculated as*

$$PICP := \frac{1}{N} \sum_{(\mathbf{x},y) \sim \mathcal{D}} \mathbb{I}_{\mu_1 \leq y \leq \mu_2}.$$

**Definition 2** (Mean Prediction Interval Width (MPIW) (Tagasovska & Lopez-Paz, 2019)). *Defined as the average interval width across the evaluation dataset, this is calculated as*

$$MPIW := \frac{1}{N} \sum_{(\mathbf{x},y) \sim \mathcal{D}} |\mu_2 - \mu_1|.$$

**Definition 3** (Width-coverage Pearson's Correlation (WCPC) (Feldman et al., 2021)). *Denoting* $\mathbf{w}$ *as the vector of interval widths where* $w_i = |\mu_2(\mathbf{x}_i) - \mu_1(\mathbf{x}_i)|$ *and* $\mathbf{m}$ *as the indicator vector of coverage events where* $m_i = \mathbb{I}_{y_i \in [\mu_1(\mathbf{x}_i), \mu_2(\mathbf{x}_i)]}$ *for* $i \in \{1, 2, \ldots, N\}$, *then we define*

$$WCPC := \left| \frac{Cov(\mathbf{w}, \mathbf{m})}{Var(\mathbf{w})Var(\mathbf{m})} \right|.$$

**Definition 4** (Hilbert-Schmidt Independence Criterion (HSIC) (Greenfeld & Shalit, 2020; Feldman et al., 2021)). *In this definition, upper case bold letters denote matrices. Using the same definitions of* $\mathbf{w}$ *and* $\mathbf{m}$ *from Definition 3, the coverage kernel matrix is given by* $\mathbf{R}_{i,j} = k(\mathbf{m}_i, \mathbf{m}_j)$ *and the width kernel is given by* $\mathbf{K}_{i,j} = k(\mathbf{w}_i, \mathbf{w}_j)$ *where* $k$ *denotes the Gaussian kernel. We also introduce a centering matrix* $\mathbf{H}_{i,j} = \delta_{i,j} - \frac{1}{N}$ *where* $\delta_{i,j} = 1$ *if* $i = j$ *and* $0$ *otherwise. Then we can calculate HSIC as*

$$HSIC := \sqrt{\frac{tr(\mathbf{KHRH})}{(N-1)^2}}$$

## E  CROSSING BOUNDS

A well-known limitation of the quantile regression approach to constructing prediction intervals is any estimation error of the population level intervals can result in *crossing bounds* where the upper bound falls *below* the lower bound (see e.g. Brando et al. (2022); Park et al. (2022)). Apart from being a conceptual limitation, this can also affect the users trust in the system. A key advantage of directly estimating the interval as proposed in this work is that the raw model outputs are not strictly associated with being a particular bound. In other words, our objective is invariant to permutations of the upper and lower bounds, and simply sets the lower bound to be the minimum and the upper bound to be the maximum of the two model outputs. Therefore, *which* output neuron acts as either bound can even change on a per-example basis. Formally, this can be observed in the $\kappa = (y - \mu_1)(y - \mu_2)$ term in our loss function (Equation (1)) which clearly invariant to permutations between $\mu_1$ and $\mu_2$. In contrast, crossing bounds in the case of quantile regression will result in miscoverage if the bounds are permuted to avoid a negative interval. This may be considered a distinct advantage of the direct interval prediction approach which sidesteps the conceptual issue of crossing quantiles which occurs due to the independent estimation of the upper and lower bounds in quantile-based approaches.

Recent work in Brando et al. (2022) has proposed methods for preventing crossing bounds in the case of quantile regression by using Chebyshev polynomials to add additional constraints into the the objective. In what follows we compare our proposed QFR-W method to this non-crossing quantile method using the same experimental setup as in Table 3. As this method is a quantile estimation method and does not prescribe how to construct intervals we include two approaches (a) "narrowest" where we bin search the narrowest interval at each new inference and (b) "symmetric" where we use the symmetric quantiles (0.05, 0.95). The results are included in Table 7.

In these results we note that, as with the previously evaluated methods for estimating all quantiles simultaneously, the non-crossing quantile approach consistently struggles to maintain coverage at test time. We seek solutions that minimize interval width *for a fixed level of coverage* which is not achieved by this baseline which we attribute to two limitations. Firstly, when we select the narrowest possible interval from a set of possible intervals that each obtain marginal coverage, it is not a random choice and it is *more* likely that we are selecting a particular interval in which coverage is not maintained. Therefore, analogous to the effect of *overfitting* in the standard point prediction setting, we would expect that picking the narrowest intervals is likely to result in more miscoverage

Table 7: **Comparison between QRF-W vs Deep Non-crossing Quantiles**. We present two different versions of the deep non-crossing quantiles method: "Narrowest" and "Symmetric" (see text for details). For the 9 datasets, we display interval width and the coverage level achieved in parentheses.

| Dataset | QRF-W | Narrowest | Symmetric |
|---------|-------|-----------|-----------|
| concrete | 0.43 (89.95) | 0.20 (57.33) | 0.30 (73.30) |
| boston | 0.51 (89.8) | 0.26 (64.90) | 0.36 (78.53) |
| naval | 0.01 (90.95) | 0.01 (97.43) | 0.02 (99.04) |
| energy | 0.13 (89.87) | 0.07 (50.39) | 0.14 (72.86) |
| wine | 0.35 (89.44) | 0.19 (62.56) | 0.28 (76.75) |
| power | 0.03 (90.73) | 0.02 (73.70) | 0.03 (89.83) |
| protein | 1.59 (90.07) | 1.13 (78.04) | 1.50 (89.26) |
| kin8nm | 0.34 (90.59) | 0.25 (70.96) | 0.38 (85.87) |
| yacht | 0.27 (93.23) | 0.29 (51.29) | 0.35 (61.61) |

events than selecting a fixed interval (as is reflected in these results). Secondly, learning all quantiles simultaneously is a more challenging learning problem than simply learning two quantiles for a fixed capacity model. This results in poorer predictive performance at the task in hand.

## F    MOTIVATION FOR MINIMIZING INTERVAL WIDTH

Although several methods might achieve a desired level of marginal coverage with their intervals, the task of finding a set of intervals that obtain such coverage is generally an underspecified problem with a potentially infinite number of admissible solutions. Given this, any interval-producing method is required to introduce some additional regularization (either implicit or explicit) in order to select among the possible solutions. In the case of quantile regression, the pair of symmetric quantiles are selected. However, given that other attributes are typically desired in these intervals (as illustrated by the large body of work that attempts to minimize interval width (Chung et al., 2021; Tagasovska & Lopez-Paz, 2019; Pearce et al., 2018) or maximize conditional coverage (Feldman et al., 2021; Hunter & Lange, 2000)), for many applications there are likely more preferable solutions than the symmetric quantiles found via quantile regression. One real-world example where this is the case is in renewable energy sources where it has been noted that "the probability distribution of the renewable energy source's power output is generally skewed, thereby the width of CPIs is often unnecessarily wide" (Zhang et al., 2023).

One additional motivation for minimizing the width of intervals beyond the advantage of narrower predictive intervals is that the minimal width intervals generally lie in denser regions of the underlying probability distribution. Because we estimate these statistics using a sample from the population distribution we typically incur some variance in this estimate which is typically lower in the more dense regions of the probability space. To illustrate this, consider the distribution we presented in Figure 1 of the main text where the ground truth intervals are known. Now suppose we take a sample from this distribution (i.e. a training set) and estimate both the empirical minimum width intervals and the empirical quantiles. In both cases there is likely to be some estimation error. If we were to repeat this process multiple times we note that the variance around the estimates in lower-density regions (where the symmetric quantiles lie) is likely to be greater than in the high-density regions (where the minimum width intervals lie). As a consequence of this, quantile estimation methods may suffer from larger errors in estimating true quantiles (particularly in small sample regimes) resulting in more difficulties in obtaining exact coverage. We verify this claim empirically in Figure 3.

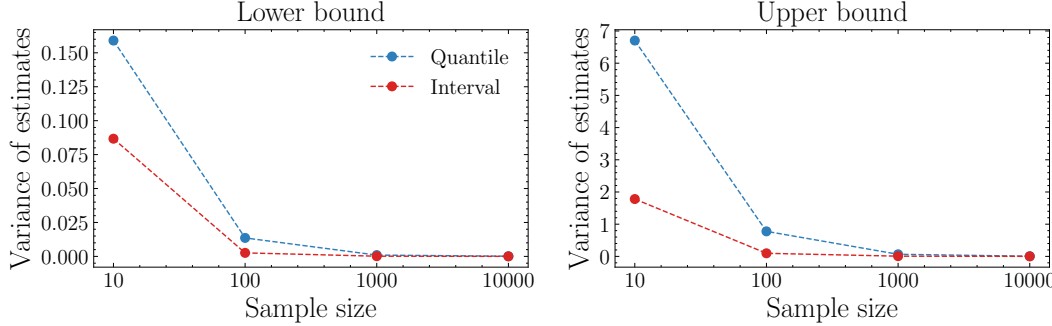

Figure 3: **Variance of the estimated bounds.** Estimating the minimal width interval rather than symmetric quantiles is also likely to result in lower variance estimates of the two bounds due to being estimated in more dense regions of the distribution. This is illustrated on the log-normal distribution example from Figure 1. We take samples of various sizes from this distribution and estimate both the quantiles and the minimum width bounds. We find that the variance of these estimates is significantly larger when estimating the former.

## G    ADDITIONAL RESULTS

### G.1    LOW COVERAGE PROBABILITY

Lower coverage intervals (i.e. narrower intervals) can lead to greater instability in the estimates. Thus, it is valuable to evaluate our method in the low coverage probability regime to ensure this increased instability doesn't have a disproportionally large effect on our method. We empirically analyze the relative performance of our method for a lower coverage level in Table 8.

Table 8: **Comparison QR vs QRF-W on 50% targeted coverage.** For the 9 datasets, we display interval width and the coverage level achieved in parentheses ± a standard error for both. All results report the test set mean over 20 runs.

| Dataset | QR | QRF-W |
|---|---|---|
| boston | 0.145 ± 0.006 (44.71 ± 1.60) | 0.509 ± 0.082 (49.56 ± 2.05) |
| concrete | 0.217 ± 0.008 (53.42 ± 1.04) | 0.136 ± 0.001 (50.49 ± 0.96) |
| energy | 0.106 ± 0.004 (53.02 ± 2.36) | 0.091 ± 0.006 (48.77 ± 2.92) |
| kin8nm | 0.132 ± 0.001 (50.05 ± 0.37) | 0.489 ± 0.019 (50.38 ± 0.43) |
| naval | 0.013 ± 0.001 (49.67 ± 2.17) | 0.002 ± 0.000 (51.42 ± 2.16) |
| power | 0.010 ± 0.000 (47.27 ± 1.57) | 0.012 ± 0.001 (48.36 ± 2.26) |
| protein | 0.829 ± 0.003 (49.69 ± 0.16) | 0.662 ± 0.005 (52.66 ± 0.35) |
| wine | 0.271 ± 0.052 (57.08 ± 7.22) | 0.152 ± 0.006 (49.59 ± 4.12) |
| yacht | 0.404 ± 0.022 (49.92 ± 2.53) | 0.156 ± 0.007 (48.71 ± 1.87) |

As we might expect, both methods have increased variance in their estimates at this level. The mean standard errors for 90% coverage rate are 0.99 and 0.57 for QFR-W and QR respectively while they are 1.9 (QFR-W) and 2.1 (QR) for the 50% coverage rate. Using the same definition of achieving coverage as in Table 3, we find that the only case of not achieving coverage is QR on Boston. Of the remaining 8 cases, QFR-W obtains narrower intervals in 6/8 cases, tying 1/8 and QR obtaining narrower intervals in 1/8. Overall this is consistent with previous results indicating that QFR-W also obtains more narrow intervals when we consider alternative coverage levels.

### G.2    NON-SMOOTHED KIN8NM INTERVALS

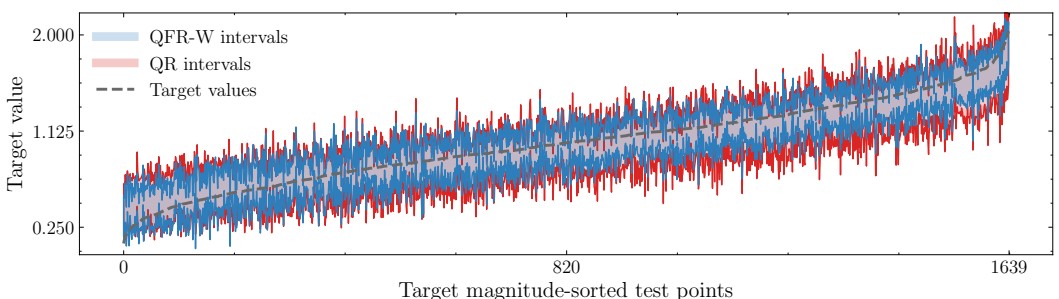

Figure 4: **Non-smoothed kin8nm intervals.** Figure 2 (left) without Savitzky–Golay filter applied. In this version it is apparent that micoverage events do occur as we would expect while the difference in interval width is less legible.

