# OpenReview forum: "Quantile-Free Regression: A Flexible Alternative to Quantile Regression"
_ICLR.cc/2024/Conference — Submitted to ICLR 2024_

### Official Review · Reviewer_LRPB · 2023-10-24

**Soundness:** 3 good
**Presentation:** 3 good
**Contribution:** 3 good
**Rating:** 5
**Confidence:** 3

**Summary:**

This paper studies loss functions for quantile regression problems. The authors propose a quantile-free regression method to estimate quantile-based intervals for output variables. Theoretical analysis reveals that the proposed loss function yields a consistent result for an infinite sample limit. Though the l_2 regularization for quantile intervals induces a bias to the estimated interval, the authors proposed how to adjust the regularization parameter and the coverage probability to maintain the consistency of the proposed method. Numerical studies indicate the proposed method outperforms some existing methods for estimating the quantile-based intervals.

**Strengths:**

- The problem considered in the paper is important. The authors proposed an interesting loss function for estimating two quantile regressions simultaneously.
- Theorem 3.2 reveals the relationship between the coverage probability and regularization parameter. This is an interesting and important result.

**Weaknesses:**

- From the definition, \mu_l(x) should be less than \mu_u(x) for arbitrary x. Is there any theoretical guarantee that the estimated upper and lower quantiles satisfy that inequality? Supplementary comments on that would be helpful for practitioners.

- Theoretical analysis is insufficient to guarantee the proposed method's effectiveness. The authors could theoretically investigate the estimation accuracy and prediction performance of the estimated intervals using the quantiles. The theoretical guarantee would be hard when a neural network model is employed. On the other hand, theoretical analysis for simple models such as the linear model would be possible.

- The stability of the estimator can depend on the coverage probability. Intuitively, the estimator for the small coverage probability will be vulnerable to a small data perturbation. Showing some theoretical or numerical explanation on that will be helpful for readers.

**Questions:**

- From the definition, \mu_l(x) should be less than \mu_u(x) for arbitrary x. Is there any theoretical guarantee that the estimated upper and lower quantiles satisfy that inequality? Supplementary comments on that would be helpful for practitioners.

- Theoretical analysis is insufficient to guarantee the proposed method's effectiveness. The authors could theoretically investigate the estimation accuracy and prediction performance of the estimated intervals using the quantiles. The theoretical guarantee would be hard when a neural network model is employed. On the other hand, theoretical analysis for simple models such as the linear model would be possible. If possible, I recommend the authors supplement an asymptotic or finite-sample analysis for the proposed estimator and compare the result to existing works.

- The stability of the estimator can depend on the coverage probability. Intuitively, the estimator for the small coverage probability will be vulnerable to a small data perturbation. Showing some theoretical or numerical explanation on that will be nice. A more detailed theoretical analysis will make this work more solid.

---

> ### Author Response · Authors · 2023-11-17
> **Rebuttal [1/2]**
>
> Thank you for your thoughtful comments and suggestions! We address each of the points raised in turn below.
>
> **_Q1 - Crossing quantiles._**
>
> We appreciate this reviewer raising the subject of theoretical guarantees around the avoidance of crossing of the upper and lower bound of the intervals as this should be considered an advantage of our approach that we have not emphasized in the text. While explicitly linking the quantiles to being specific upper and lower bounds is necessary in quantile regression (due to the requirement of estimating quantiles), this is not the case for our method which directly estimates intervals. Our proposed objective is invariant to permutations of the upper and lower bounds, and simply sets the lower bound to be the minimum and the upper bound to be the maximum of the two values. Therefore, _which_ output neuron acts as either bound can even change on a per-example basis. Formally, this can be observed in the $\kappa = (y - \mu_l)(y - \mu_u)$ term in our loss function which is clearly invariant to permutations between $\mu_l$ and $\mu_u$. We appreciate that this may have been obfuscated by our notation using subscripts $l$ and $u$ which we have now changed to better reflect that neither of these values are explicitly tied to a particular bound. We consider this to be a distinct advantage of our direct interval prediction approach which sidesteps the conceptual issue of crossing quantiles which occurs due to the independent estimation of the upper and lower bounds in quantile-based approaches.
>
> **Action taken**: We have highlighted our advantages in terms of non-crossing intervals in the text while updating our notation to reflect this.
>
> **_Q2 - Additional theoretical analysis._**
>
> We agree that further theoretical analysis of the behavior of our estimator would be a valuable addition to this work. We have derived a new theorem for our proposed QFR loss $\mathcal{L}^{QFR}$ demonstrating that the interval resulting from the minimization of the loss directly covers the right proportion of samples that correspond to the targeted coverage, regardless of the size of the sample (for ease of access, we provide the new proof at [this link](https://imgur.com/a/LacyCzo)). This property is called the “quantile property” when true for a quantile estimator loss function [1] and illustrates that our proposed expression is well motivated. This reassuring property reinforces our claim that this expression provides a suitable objective for the goal of generating well-calibrated prediction intervals in the finite sample setting.
>
> **Action taken**: We have added this new theorem to the manuscript.

---

> > ### Author Response · Authors · 2023-11-17
> > **Rebuttal [2/2]**
> >
> > **_Q3 - Lower coverage probability._**
> >
> > It is indeed true that lower coverage intervals (i.e. narrower intervals) can lead to greater instability in the estimates. We agree that it is valuable to evaluate our method in this regime to ensure this increased instability doesn’t have a disproportionally large effect on our method. Therefore we empirically analyse the relative performance of our method for a lower coverage level in a new experiment. Below we include equivalent results to Table 3 from our main text evaluated at the 50% coverage level instead. We include interval width and the coverage level achieved in parentheses $\pm$ a standard error for both.
> >
> >
> > |  Dataset  |                      	QR                       	|              	QRF-W               	|
> > |---------|--------------------------------------------------------|-------------------------------------------|
> > | concrete| 0.217 ± 0.008 (53.42 ± 1.04)                      	| 0.136 ± 0.001 (50.49 ± 0.96)          	|
> > | power   | 0.010 ± 0.000 (47.27 ± 1.57)                      	| 0.012 ± 0.001 (48.36 ± 2.26)          	|
> > | wine	| 0.271 ± 0.052 (57.08 ± 7.22)                      	| 0.152 ± 0.006 (49.59 ± 4.12)          	|
> > | yacht   | 0.404 ± 0.022 (49.92 ± 2.53)                      	| 0.156 ± 0.007 (48.71 ± 1.87)          	|
> > | naval   | 0.013 ± 0.001 (49.67 ± 2.17)                      	| 0.002 ± 0.000 (51.42 ± 2.16)          	|
> > | energy  | 0.106 ± 0.004 (53.02 ± 2.36)                      	| 0.091 ± 0.006 (48.77 ± 2.92)          	|
> > | Boston  | 0.145 ± 0.006 (44.71 ± 1.60)                      	| 0.509 ± 0.082 (49.56 ± 2.05)          	|
> > | kin8nm  | 0.132 ± 0.001 (50.05 ± 0.37)                      	| 0.489 ± 0.019 (50.38 ± 0.43)          	|
> > | protein | 0.829 ± 0.003 (49.69 ± 0.16)                      	| 0.662 ± 0.005 (52.66 ± 0.35)          	|
> >
> > As expected, both methods have increased variance in their estimates at this level. The mean standard errors for 90% coverage rate are 0.99 and 0.57 for QFR-W and QR respectively while they are 1.9 (QFR-W) and 2.1 (QR) for the 50% coverage rate. Using the same definition of achieving coverage as included in the main text, we find that the only case of not achieving coverage is QR on Boston. Of the remaining 8 cases, QFR-W obtains narrower intervals in 6/8 cases, tying 1/8 and QR obtaining narrower intervals in 1/8. Overall this is consistent with previous results indicating that QFR-W also obtains more narrow intervals when we consider alternative coverage levels.
> >
> > As a related point, we note that improved stability of the estimates may also be considered an advantage for using minimal width bounds over symmetric quantiles more generally. Since minimum width bounds will generally be found in denser regions of the underlying distribution than symmetric quantiles, the variance of their estimates will tend to be lower. We illustrate this effect on the skewed distribution presented in Figure 1 ([see this link for results](https://imgur.com/a/t9V4txS)). This added stability can result in a more consistent realization of the desired coverage level on test data.
> >
> > **Action taken**: We have added these results extending our analysis to a new coverage target to the main text. We have also added some discussion on the stability of the bound estimates including the linked results.
> >
> >
> > [1] Takeuchi, Ichiro, et al. "Nonparametric quantile estimation." (2006).

---

> > ### Comment · Reviewer_LRPB · 2023-11-23
> >
> > Thank you for your responses and revisions.
> >
> > As a new theoretical analysis, I realized that Theorem 3.2 (QFR with finite samples) was added. This is an interesting result. However, I do not think the theorem reveals an inferential statistical property of the proposed estimator, though it may provide a descriptive statistical property. The authors could evaluate the statistical accuracy of the proposed method in an asymptotic or finite-sample scenario.

---

> > > ### Author Response · Authors · 2023-11-23
> > >
> > > Thank you for your response!
> > >
> > > Ah, we understand the point you are making and agree that this is a nice idea. We have derived a new result based on your suggestion that uses our Theorem 3.2 to show that the asymptotic variance is bounded using Monte Carlo theory. You can find the result by clicking [the following link](https://imgur.com/a/Cm9et6X) which we will add to the next version of the paper.
> > >
> > > These theoretical results of our estimators provide some nice properties about our proposed object, reinforcing our claim that this is a well-motivated approach to the task of constructing prediction intervals. We wish to also highlight that our method holds similar or superior theoretical guarantees compared to what is reported by the baseline methods. This is confirmed by our empirical experiments demonstrating excellent performance relative to the baselines in the finite sample regime.

---

### Official Review · Reviewer_eWa2 · 2023-10-29

**Soundness:** 2 fair
**Presentation:** 3 good
**Contribution:** 2 fair
**Rating:** 3
**Confidence:** 4

**Summary:**

The paper proposes a new loss function that directly predicts the interval and frees the need to specify upper and lower quantile levels when predicting the conditional confidence intervals. The rationale behind this construction is to write out the optimization problem (with the vanilla pinball loss) for the upper and lower quantiles respectively and then combine these two problems in a way such that the exact value of the lower quantile level is eliminated, and only the difference between upper and lower quantile levels remains.

**Strengths:**

The paper proposes a new loss function -- the quantile-free regression (QRF) loss. The loss function is natural and interesting, especially the result in Theorem 3.2 which explicitly depicts the "confidence shrinking" effect of the regularization term. What's more, this loss is intrinsically open to the continuous search for the sharpest confidence interval, unlike traditional methods that often take a grid search among discrete choices.

**Weaknesses:**

1. The paper doesn't provide any guarantees other than the asymptotic consistency result for the proposed QRF loss. In order to argue for the advantage of this QRF loss over multiple alternatives (like, explicitly fixing the lower quantiles and upper quantiles), finite sample error would be more desirable.

2. It makes sense that when the conditional distribution is skewed, using a symmetric confidence interval is not an optimal choice, yet the missing part of the argument is, why the QRF loss is an optimal or better choice? Say, under what distribution shape will the QRF be preferable? The current theoretical evidence only supports that QRF is consistent in a marginal sense (which is weak), but it doesn't show the advantage of the QRF over other methods as claimed. For the numerical experiments, the synthetic experiment is a bit special, while the real-data experiments didn't include many other calibration methods, but only those quantile regression-based ones.

3. The loss function seems to add an additional non-convexity because of the cross term of the product $\mu_l\cdot \mu_u$. I think the common belief is that we need to be very cautious when we introduce non-convexity to the loss function.

**Questions:**

See above.

---

> ### Author Response · Authors · 2023-11-17
> **Rebuttal [1/2]**
>
> Thank you for your thoughtful comments and suggestions! We address each of the points raised in turn below.
>
> **_Q1 - Non-asymptotic guarantees_**
>
> We agree that some non-asymptotic guarantees on the behavior of our estimator would be a valuable addition to this work. We have derived a new theorem for our proposed QFR loss, $\mathcal{L}^{QFR}$, demonstrating that the interval resulting from the minimization of the loss directly covers the right proportion of samples that correspond to the targeted coverage, regardless of the size of the sample (for ease of access, we provide the new proof at [this link](https://imgur.com/a/LacyCzo)). This property is called the “quantile property” when true for a quantile estimator loss function [8], and illustrates that our proposed expression is well motivated. This reassuring property reinforces our claim that this expression provides a suitable objective for the goal of generating well-calibrated prediction intervals in the finite sample setting.
>
> **Action taken**: We have added this new theorem to the manuscript.
>
> **_Q2 - Why is QFR preferable?_**
>
> We thank the reviewer for raising this point as it is something that should have been made more clear in the text. The key idea of our work is that although several methods might achieve a desired level of marginal coverage with their intervals, this is an underspecified problem with potentially an infinite number of admissible solutions. Given this, any interval-producing method is required to introduce some additional regularization (either implicit or explicit) in order to select among the possible solutions. In the case of quantile regression, the pair of symmetric quantiles are selected. However, given that other attributes are typically desired in these intervals (as illustrated by the large body of work that attempts to minimize interval width [1-3] or maximize conditional coverage [4,5]), there are likely more preferable solutions than quantile regression. One real-world example where this is the case is in renewable energy sources where it has been noted that “the probability distribution of the renewable energy source’s power output is generally skewed, thereby the width of CPIs is often unnecessarily wide” [6].
>
> In our work, we ensure this decision is made explicitly by the user such that the most appropriate intervals can be found for a given application by using the corresponding regularization term. Our experiments in Tables 3 & 4 then illustrate that our objective results in intervals that better achieve these properties when evaluated on standard benchmarks. For Table 3 this improvement is not surprising as symmetric intervals are generally not the most narrow possible intervals (because real-world distributions are unlikely to be perfectly symmetric). For Table 4 this improvement is also expected as it provides greater flexibility on the solution to select one that achieves greater independence between width and miscoverage events.
>
> We appreciate that the uncertainty quantification literature is vast and there exist many differing approaches. In Appendix A we provided an extended discussion on this literature which highlighted that _single model approaches_ such as quantile regression have distinct advantages that make research in this area valuable in its own right. Factors such as their minimal computational overhead and lack of assumptions on the data have resulted in the extensive application of these methods in practice (see e.g. [7] for an overview of just some of the vast number of real-world implementations of this method). We believe it would fall outside the scope of this conference paper to survey and compare the various categories across all of the uncertainty quantification literature while accounting for their relative strengths and weaknesses to the appropriate rigorous standard required for a conference such as ICLR.
>
> **Action taken**: We have updated our text to better reflect this motivation.

---

> > ### Author Response · Authors · 2023-11-17
> > **Rebuttal [2/2]**
> >
> > **_Q3 - Non-convex loss._**
> >
> > We completely agree that convexity is conceptually a nice property to have in a loss function. However, the theoretical guarantees afforded to convex loss functions (e.g. converging to the global minimum) typically require additional assumptions which are generally not met in the deep learning setting anyway. Therefore, we would argue that methods should be judged on their empirical performance which, in the case of our proposed method, is generally superior to existing convex baselines (indeed, neural networks themselves are typically non-convex which has certainly not limited their value!).
> >
> >
> >
> > [1] Chung, Youngseog, et al. "Beyond pinball loss: Quantile methods for calibrated uncertainty quantification." Advances in Neural Information Processing Systems 34 (2021): 10971-10984.
> >
> > [2] Tagasovska, Natasa, and David Lopez-Paz. "Single-model uncertainties for deep learning." Advances in Neural Information Processing Systems 32 (2019).
> >
> > [3] Pearce, Tim, et al. "High-quality prediction intervals for deep learning: A distribution-free, ensembled approach." International conference on machine learning. PMLR, 2018.
> >
> > [4] Feldman, Shai, Stephen Bates, and Yaniv Romano. "Improving conditional coverage via orthogonal quantile regression." Advances in neural information processing systems 34 (2021): 2060-2071.
> >
> > [5] Hunter, David R., and Kenneth Lange. "Quantile regression via an MM algorithm." Journal of Computational and Graphical Statistics 9.1 (2000): 60-77.
> >
> > [6] Zhang, Yufan, Honglin Wen, and Qiuwei Wu. "A contextual bandit approach for value-oriented prediction interval forecasting." IEEE Transactions on Smart Grid (2023).
> >
> > [7] Huang, Qi, et al. "Quantile regression models and their applications: a review." Journal of Biometrics & Biostatistics 8.3 (2017): 1-6.
> >
> > [8] Takeuchi, Ichiro, et al. "Nonparametric quantile estimation." (2006).

---

### Official Review · Reviewer_iRaW · 2023-10-30

**Soundness:** 2 fair
**Presentation:** 3 good
**Contribution:** 2 fair
**Rating:** 3
**Confidence:** 4

**Summary:**

This paper introduces Quantile-Free Regression (QFR). QFR targets to compute an upper and lower interval bounds such that the interval achieves a user-specified target coverage level. Since there are infinitely many solutions to the unregularized QFR problem, the authors proposed two regularization terms to induce some preference among solutions. The two regularization terms favor small interval width and independence between interval width and coverage level, respectively. Theoretical analysis guarantees that the trained QFR model covers exactly the target level. Empirical results show that the QFR model and the two regularization terms can induce the desired properties.

**Strengths:**

- The paper proposes a simple approach to compute two quantile estimates such that they cover a target level of data. A novel objective function for performing QFR is proposed.
- Existing interval regression methods do not guarantee asymptotic coverage, while the proposed approach has theoretical guarantee and is empirically tested.
- This paper is clearly written and very easy to follow.
- An extensive literature review which discusses the connections and differences of the proposed work and existing approaches is given.
- Extensive synthetic and real experimental data are used to test the performance of the model and verify the claims made by the authors.

**Weaknesses:**

- In the experiments, only the case of high-level coverage (90%) is tested. More extensive experiments considering low and mid-level coverage should be performed. It seems that there is no guarantee on monotonicity, i.e., \mu_l <= \mu_u. I wonder if monotonicity is preserved for a low-level target coverage level
- The comparing methods are mostly basic models. There are SQR approaches which compute the estimates over all quantile levels and has comparable computational cost to QR approaches, e.g., the monotonic regression splines approach and [1]. The authors can consider adding more methods to the experiments.
- I am not sure about the statement "QFR maintains QR's strengths". While the proposed method can flexibly achieve the target coverage level, the corresponding quantile levels for the interval cannot be obtained. The main strength of QR is that the estimates for specific quantile levels are provided.


[1] Park, Y., Maddix, D., Aubet, F. X., Kan, K., Gasthaus, J., & Wang, Y. (2022, May). Learning quantile functions without quantile crossing for distribution-free time series forecasting. In International Conference on Artificial Intelligence and Statistics (pp. 8127-8150). PMLR.

**Questions:**

- My main concern is about the model performance on other levels of coverage. How does the model perform compared to the baselines, when \alpha is say, 30%, 50% and 70%?
- The practical advantages of narrow interval width and independence between interval width and coverage are unclear. For instance, is there any real-world situation where the property shown in figure 1 is desirable?
- SQF computes the quantile estimates for more than two values. In the experimental results, how is the coverage of SQF computed?
- Is there any intuition on why QFR outperforms QR on getting a 90% coverage?

---

> ### Author Response · Authors · 2023-11-16
> **Rebuttal [1/4]**
>
> Thank you for your thoughtful comments and suggestions! We address each of the points raised in turn below.
>
> **_Q1 - Coverage levels and monotonicity_**
>
> We appreciate the suggestion to consider target coverage levels beyond the 90% already included in the text. We focused on 90% coverage as this is the standard level used in previous works throughout the literature [1-3]. However, it is indeed true that different applications may require different coverage levels and, therefore, investigating the impact of adjusting this value may be valuable to future readers. Below we include equivalent results to Table 3 from our main text evaluated at the 50% coverage level instead. We include interval width and the coverage level achieved in parentheses $\pm$ a standard error for both.
>
> |  Dataset  |                      	QR                       	|              	QRF-W               	|
> |---------|--------------------------------------------------------|-------------------------------------------|
> | concrete| 0.217 ± 0.008 (53.42 ± 1.04)                      	| 0.136 ± 0.001 (50.49 ± 0.96)          	|
> | power   | 0.010 ± 0.000 (47.27 ± 1.57)                      	| 0.012 ± 0.001 (48.36 ± 2.26)          	|
> | wine	| 0.271 ± 0.052 (57.08 ± 7.22)                      	| 0.152 ± 0.006 (49.59 ± 4.12)          	|
> | yacht   | 0.404 ± 0.022 (49.92 ± 2.53)                      	| 0.156 ± 0.007 (48.71 ± 1.87)          	|
> | naval   | 0.013 ± 0.001 (49.67 ± 2.17)                      	| 0.002 ± 0.000 (51.42 ± 2.16)          	|
> | energy  | 0.106 ± 0.004 (53.02 ± 2.36)                      	| 0.091 ± 0.006 (48.77 ± 2.92)          	|
> | Boston  | 0.145 ± 0.006 (44.71 ± 1.60)                      	| 0.509 ± 0.082 (49.56 ± 2.05)          	|
> | kin8nm  | 0.132 ± 0.001 (50.05 ± 0.37)                      	| 0.489 ± 0.019 (50.38 ± 0.43)          	|
> | protein | 0.829 ± 0.003 (49.69 ± 0.16)                      	| 0.662 ± 0.005 (52.66 ± 0.35)          	|
>
> Interestingly, both methods have increased variance in their estimates at this level. It appears that as coverage decreases the intervals become narrower and, just as we would expect as approaching point estimates, the variance of the miscoverage rate increases. The mean standard errors for 90% coverage rate are 0.99 and 0.57 for QFR-W and QR respectively while they are 1.9 (QFR-W) and 2.1 (QR) for the 50% coverage rate. Using the same definition of achieving coverage as included in the main text, we find that the only case of not achieving coverage is QR on Boston. Of the remaining 8 cases, QFR-W obtains narrower intervals in 6/8 cases, tying 1/8 and QR obtains narrower intervals in 1/8. Overall this is consistent with previous results indicating that QFR-W also obtains more narrow intervals when we consider alternative coverage levels.
>
> We appreciate this reviewer raising the subject of monotonicity of the intervals as this should be considered an advantage of our approach that we have not emphasized in the text. While explicitly linking the quantiles to being specific upper and lower bounds is necessary in quantile regression (due to the requirement of estimating quantiles), this is not the case for our method which directly estimates intervals. Our proposed objective is invariant to permutations of the upper and lower bounds, and simply sets the lower bound to be the minimum and the upper bound to be the maximum of the two values. Therefore, _which_ output neuron acts as either bound can even change on a per-example basis. Formally, this can be observed in the $\kappa = (y - \mu_l)(y - \mu_u)$ term in our loss function which is clearly invariant to permutations between $\mu_l$ and $\mu_u$. We appreciate that this may have been obfuscated by our notation using subscripts $l$ and $u$ which we have now changed to better reflect that neither of these values are explicitly tied to a particular bound.
>
> **Action taken**: We have added these results extending our analysis to a new coverage target to the main text. We have also highlighted our advantages in terms of monotonicity in the text while updating our notation to reflect this.

---

> > ### Author Response · Authors · 2023-11-16
> > **Rebuttal [2/4]**
> >
> > **_Q2 - Additional baseline._**
> >
> > We thank this reviewer for suggesting we compare to baselines that consider crossing quantiles. The suggested paper by Park et. al. is concerned with non-crossing quantiles in the time series setting. As our experiments are performed on static datasets, we instead implement the method of [4] which proposes a similar non-crossing quantile method for static data. We evaluate this method as an additional baseline with the same setup as our Table 3. As this method is a quantile estimation method and does not prescribe how to construct intervals we include two approaches (a) “narrowest” where we bin search the narrowest interval at each new inference and (b) “symmetric” where we use the symmetric quantiles (0.05, 0.95). The results are included below.
> >
> > | Dataset  |  QFR-W | Narrowest | Symmetric  |
> > |---|---|---|---|
> > | Concrete  | 0.43 (89.95)  | 0.20 (57.33) | 0.3 (73.30)  |
> > | Boston  | 0.51 (89.8)  | 0.26 (64.90)  | 0.36 (78.53)  |
> > | Naval  | 0.01 (90.95)  | 0.01 (97.43)  | 0.02 (99.04)  |
> > | Energy | 0.13 (89.87)  | 0.07 (50.39)  | 0.14 (72.86)  |
> > | Wine  |  0.35 (89.44) |  0.19 (62.56) | 0.28 (76.75)  |
> > | Power  |  0.03 (90.73) | 0.02 (73.70)  |  0.03 (89.83) |
> > | Protein  | 1.59 (90.07)  | 1.13 (78.04)  | 1.50 (89.26)  |
> > | Kin8nm  | 0.34 (90.59)  | 0.25 (70.96)  | 0.38 (85.87)  |
> > | Yacht  | 0.27 (93.23)  |  0.29 (51.29) | 0.35 (61.61)  |
> >
> > We note that, as with the previously evaluated methods for estimating all quantiles simultaneously, this approach consistently struggles to maintain coverage at test time. We seek solutions that minimize interval width _for a fixed level of coverage_ which is not achieved by this baseline which we attribute to two limitations. Firstly, when we select the narrowest possible interval from a set of possible intervals that each obtain marginal coverage, it is not a random choice and it is _more_ likely that we are selecting a particular interval in which coverage is not maintained. Therefore, we would expect that picking the narrowest intervals is likely to result in more miscoverage events than selecting a fixed interval (as is reflected in these results). Secondly, learning all quantiles simultaneously is a more challenging learning problem than simply learning two quantiles for a fixed capacity model. This results in poorer predictive performance at the task in hand.
> >
> > **Action taken**: We have added these results to the text and extended our discussion on the limitations of estimating all quantiles simultaneously.
> >
> >
> > **_Q3 - Phrasing of comparison._**
> >
> > We appreciate this reviewer pointing out that our phrasing of the statement in our abstract could be misinterpreted as implying that we also estimate quantiles. The complete statement was “In this work, we propose Quantile-Free Regression (QFR), a direct replacement for quantile regression which liberates it from this limitation whilst maintaining its strengths”.
> >
> > **Action taken**: We have updated this sentence to avoid confusion. It now states: “In this work, we propose Quantile-Free Regression (QFR), a direct replacement for quantile regression which liberates it from this limitation whilst maintaining its ability to generate reliable prediction intervals”.

---

> > > ### Author Response · Authors · 2023-11-16
> > > **Rebuttal [3/4]**
> > >
> > > **_Q4 - Practical advantages._**
> > >
> > > We are happy to provide further context on the practical advantages of both narrow intervals or independence between interval width and coverage. We note that the generally accepted view throughout the uncertainty quantification literature is that although several methods might achieve a desired level of marginal coverage with their intervals, this is an underspecified problem with a potentially infinite number of admissible solutions. Therefore, previous works seek to compare solutions based on additional desirable qualities in those intervals.
> > >
> > > * *Interval width* - Probably the most popular example of such a quality is interval width which is evaluated ubiquitously (see e.g. the baseline methods to which we compare [2,3,5]). The idea is simply that the more narrow the region in which we have $\alpha$% confidence that the target lies, the more informative the intervals are for guiding the practitioner on what the final point value will be. The value of narrower intervals for decision-makers has been expressed more formally in previous works such as [6], which explicitly models the tradeoff between the _disutility_ of wider intervals against the _disutility_ of greater miscoverage rates for decision-makers. One example where minimizing interval width is of value is in financial applications where a prediction interval might be compared to some threshold value for a downstream decision (e.g. if a prediction is with $\alpha$ probability below a threshold it is a profitable buy/sell) where more fine-grained prediction intervals can allow for a higher quantity of profitable actions to be taken (see e.g. portfolio selection [7]).
> > >
> > > * *Independence between interval width and coverage* - When a model produces predictive intervals they are typically evaluated using their marginal coverage over some testing examples (i.e. the average coverage across the test set). However, an ideal solution would provide a stronger _conditional_ coverage (i.e. for any given test example, the conditional probability that the true value lies within the provided interval is $\alpha$). In almost any conceivable use case of predictive intervals this latter form of coverage would be preferable. However, reliably obtaining true conditional coverage is impossible in practice [see e.g. 8]. As an alternative, many works consider ways of reducing the bias of miscoverage events by encouraging independence between interval width and cases of miscoverage a popular approach [9,10]. For an example of a specific application (galaxy redshift estimation) where improved conditional coverage is valuable see [11].
> > >
> > > Given the underspecified nature of producing prediction intervals with a given level of coverage, any interval-producing method is required to introduce some additional regularization (either implicit or explicit) in order to select among the possible solutions. In our work, we ensure this decision is made explicitly by the user such that the most appropriate intervals can be found for a given application.
> > >
> > > **Action taken**: We have updated the manuscript to better reflect (a) the value of each of these objectives and (b) the value of providing practitioners with the flexibility to choose whichever interval properties provide the most utility.
> > >
> > >
> > > **_Q5 - Computation of SQF (sic) coverage._**
> > >
> > > Thank you for pointing out that we did not explicitly state this in the text. For SQR, we followed the method prescribed by the authors which consists of selecting the symmetric (0.05, 0.95) intervals.
> > >
> > > **Action taken**: We have added this detail to the text.

---

> ### Author Response · Authors · 2023-11-16
> **Rebuttal [4/4]**
>
> **_Q6 - QFR outperforming QR at achieving coverage?_**
>
> This is an astute observation and not something we explicitly addressed in the paper! We have two hypotheses why this might be the case.
>
> The first is simply that quantile regression introduces two sources of error (i.e. estimating each of the two quantiles) in which an error on either estimate can result in miscoverage. This is not the case for methods that estimate a single quantity (i.e. the interval directly). Note that we are not the first to notice this fact, see e.g. Sec. 2 of [12] for a much earlier work that has commented on this limitation.
>
> The second is that miscoverage may also be due to a reduction in the variance of estimation of statistical quantities in high-density regions of a probability distribution. To illustrate this, consider the distribution we presented in Fig 1 of the main text where the ground truth intervals are shown. Now suppose we take a sample from this distribution (i.e. a training set) and estimate both the empirical minimum width intervals and the empirical quantiles. In both cases, there is likely to be some estimation error. If we were to repeat this process multiple times we note that the variance around the estimates in lower-density regions (where the quantiles lie) is likely to be greater than in the high-density regions (where the minimum width intervals lie). As a consequence of this, quantile estimation methods may suffer from larger errors in estimating true quantiles (particularly in small sample regimes) resulting in more difficulties in obtaining exact coverage.
>
> Edit: We have now added an experiment in which we verify that the variance in the estimation of the bounds is indeed greater in the lower-density regions generally estimated by quantiles as described previously. We illustrate this effect on the skewed distribution presented in Figure 1 ([see this link for results](https://imgur.com/a/t9V4txS)). This result has now been added to the main text.
>
> **Action taken**: We have added a note reflecting this in the text.
>
>
>
>
> **References**
>
> [1] Koenker, Roger, and Kevin F. Hallock. "Quantile regression." Journal of economic perspectives 15.4 (2001): 143-156.
>
> [2] Chung, Youngseog, et al. "Beyond pinball loss: Quantile methods for calibrated uncertainty quantification." Advances in Neural Information Processing Systems 34 (2021): 10971-10984.
>
> [3] Tagasovska, Natasa, and David Lopez-Paz. "Single-model uncertainties for deep learning." Advances in Neural Information Processing Systems 32 (2019).
>
> [4] Brando, Axel, et al. "Deep non-crossing quantiles through the partial derivative." International Conference on Artificial Intelligence and Statistics. PMLR, 2022.
>
> [5] Pearce, Tim, et al. "High-quality prediction intervals for deep learning: A distribution-free, ensembled approach." International conference on machine learning. PMLR, 2018.
>
> [6] Landon, Joshua, and Nozer D. Singpurwalla. "Choosing a coverage probability for prediction intervals." The American Statistician 62.2 (2008): 120-124.
>
> [7] Huang, Shih-Feng, and Hsiang-Ling Hsu. "Prediction intervals for time series and their applications to portfolio selection." REVSTAT-Statistical Journal 18.1 (2020): 131-151.
>
> [8] Vovk, Vladimir. "Conditional validity of inductive conformal predictors." Asian conference on machine learning. PMLR, 2012.
>
> [9] Feldman, Shai, Stephen Bates, and Yaniv Romano. "Improving conditional coverage via orthogonal quantile regression." Advances in neural information processing systems 34 (2021): 2060-2071.
>
> [10] Hunter, David R., and Kenneth Lange. "Quantile regression via an MM algorithm." Journal of Computational and Graphical Statistics 9.1 (2000): 60-77.
>
> [11] Dey, Biprateep, et al. "Calibrated predictive distributions via diagnostics for conditional coverage." arXiv preprint arXiv:2205.14568 (2022).
>
> [12] Takeuchi, Ichiro, et al. "Nonparametric quantile estimation." (2006).

---

### Official Review · Reviewer_ccij · 2023-11-02

**Soundness:** 3 good
**Presentation:** 2 fair
**Contribution:** 4 excellent
**Rating:** 5
**Confidence:** 4

**Summary:**

This paper proposes a quantile-free regression to provide prediction intervals. Traditionally, prediction intervals are constructed by fitting quantile regressions. However, using the proposed approach, the width of the intervals can be effectively narrowed down by exploiting the asymmetries of the distributions. Overall, this paper solves a fundamental problem in machine learning and statistics.

**Strengths:**

1. The paper studies an important problem and has real potential.
2. Theoretical justification is provided via asymptotic coverage guarantees.

**Weaknesses:**

In my opinion, the main weakness is that the improvement over the existing approaches seems to be unclear. See details in the Questions Section below.

**Questions:**

1. The improvement over the existing approaches is basically marginal.
  - More importantly, the improvement in the width of the intervals might be due to the regularization term on the interval width. It seems that the same regularization can be naturally applied to QR and WS. Hence, the comparison may not be fair.
  - The above statement can also be seen from Table 2. For the skewed distribution, the proposed QFR fails to outperform QR; only the QFR-W with the regularization outperforms QR.
  - It does not make sense to strikethrough the case when the coverage probability is higher than the nominal value. For example, it seems to me that, in Table 3, QR performs much better than the proposed method in yacht data, as the width is roughly the same while the coverage of QR is much higher with a smaller standard error.

2. The essential idea of this paper, as well as some competing methods such as SQR, WS, and IR, is to put different weights on the different sides of the intervals. For example, if a distribution is skewed to the right, it is desired to include more data points on the left of the median in the interval to reduce the width. However, as shown in Figure 2, the proposed QFR-W intervals are narrower than the QR intervals on both sides. Can the author explain why this happens?

3. The name of the proposed method is slightly confusing. The new method basically uses a new loss to estimate the quantiles, so it is not really quantile-free. Also, the main purpose of the quantile regression is to estimate quantiles, and constructing intervals is typically seen as a byproduct. Can the proposed method directly be used to estimate quantiles? If not, it is not appropriate to claim the method is a direct replacement for quantile regression.

---

> ### Author Response · Authors · 2023-11-18
> **Rebuttal [1/2]**
>
> Thank you for your thoughtful comments and suggestions! We address each of the points raised in turn below.
>
> **_Q1 - Gains over baselines_**
>
> There is a slight misunderstanding here on the goal of our method. We have updated our text to further clarify our writing as this is the fundamental point of our method which we don’t want future readers to miss. We will also clarify in our rebuttal next.
>
> This reviewer has correctly noted that our core loss function, $\mathcal{L}^{QFR}$ does achieve desired coverage _but_ only performs well at minimizing width when the additional regularization term (resulting in QFR-W) is used in tandem. The separation of the _coverage achieving term_ and the _width minimizing term_ into two separate expressions is the key point of our method (we were actually attempting to highlight this in Table 2!). This is because achieving coverage in finite samples is an underspedified problem with potentially an infinite number of admissible solutions. Therefore, any interval construction method is required to introduce some additional regularization (either implicit or explicit) in order to select among the possible solutions. In our work, we ensure this decision is made explicitly by the user such that the most appropriate intervals can be found for a given application. This is in contrast to QR which simply selects the symmetric quantiles and therefore produces wider than necessary intervals. This has already been noted as a limitation of quantile-based methods in practice on real-world applications (e.g. in predicting energy consumption, research has noted “the probability distribution of the renewable energy source’s power output is generally skewed, thereby the width of CPIs is often unnecessarily wide” [1]).
>
> In fact the baseline methods typically _already attempt to minimize interval width_ (note that e.g. [2-4] all evaluate their method based on the resulting interval widths which are somewhat reduced via various means). However, despite these efforts, all quantile-based methods are fundamentally limited by either (a) choosing symmetric quantiles resulting in unnecessary excessive width or (b) estimating more quantiles than necessary resulting in a significantly more difficult learning problem. This is the key motivation for our method which discards the intermediate step of estimating quantiles and estimates the intervals directly. We have included an extended exposition of this point in Section 3.1 of the main text.
>
> Regarding the improvement in interval width over quantile regression we appreciate that the raw figures may not immediately suggest strong gains but we note that the reduction in interval width is up to 27.7% which is certainly a worthwhile improvement. We also note that our method statistically significantly outperforms quantile regression on 6/9 datasets while tying on the remaining 3. We would also point toward our conditional coverage results in Table 4 where the gains of QFR are compelling. Quantile regression is a very strong baseline and outperforming it by any margin is a result we believe to be useful for practitioners.
>
> We agree that the fact that several baselines fail to maintain the desired coverage level on test data to be notable. Although discounting results that fail to achieve coverage is standard in the literature (see e.g. [3]), we also agree that this is something we should have discussed further in the text as it reflects an important dimension on which methods should be assessed. We note that all methods were trained to obtain the same coverage level, therefore failing to achieve the desired coverage level at test time is (a) not something that we can control and (b) reflects a consistent advantage of our proposed method.
>
> Regarding the example on the dataset “yacht”, we highlight that the goal it to produce the narrowest intervals that contain 90% of test examples which is marginally better achieved by our method. While obtaining empirical coverage that is _greater than_ the desired coverage level may often be less harmful than obtaining _less than_ the desired coverage level, at a minimum this still reflects an inefficiency. Additionally, there are many applications in which we are primarily interested in the miscoverage cases (e.g. extreme events) and, therefore, overcoverage may be problematic in addition to being inefficient. We note that finding a set of intervals that covers 96.37% of test cases (when 90% was the target) still illustrates a failure in estimating the _correct quantiles_ as is the stated goal of QR.
>
> **Action taken**: We have updated the text to better highlight the benefits of our distinction between the coverage and regularization term. We have also added some discussion emphasizing the goal of achieving target coverage and the distinction between over and undercoverage.

---

> ### Author Response · Authors · 2023-11-18
> **Rebuttal [2/2]**
>
> **_Q2 - Figure 2 intervals._**
>
> The reviewer is absolutely correct in pointing out that the target values should not lie entirely within the intervals if they are achieving <100% coverage. The intention of this figure was to illustrate that QFR-W produces generally narrower intervals than QR by comparing their intervals on the test set sorted by target values. For clarity of illustration, we applied a consistent smoothing to both methods and included the unsmoothed version in the Appendix (see Fig. 3) which we referenced in the text. We note that in the unsmoothed version of this plot it is clear that there are several cases of miscoverage, consistent with both methods achieving 90% coverage as expected. However, the key point holds in both cases that _QFR-W finds narrower intervals in aggregate_.
>
> **_Q3 - Method name and purpose._**
>
> We appreciate the reviewer's feedback on the name of the method. We will spend some time considering alternatives for a final version of this paper.
>
> “The new method basically uses a new loss to estimate the quantiles, so it is not really quantile-free” - This is not quite true but we appreciate this is a subtle point and the reviewer's general sentiment (that we should make a clearer distinction between our method and quantile based methods) may still be correct. Existing methods first estimate quantiles from data and then use those quantiles to construct an interval. Our method never explicitly estimates quantiles and simply finds intervals that achieve coverage. Indeed every interval must correspond to _some_ quantile, but the corresponding quantile could change for every data point in the dataset. Therefore, these intervals are not associated with any particular quantiles in a meaningful way.
>
> “the main purpose of the quantile regression is to estimate quantiles, and constructing intervals is typically seen as a byproduct” - It is correct that the original Koenker & Bassett (1978) introduction of quantile regression was primarily interested in the task of estimating quantiles. Recent works have generally referred to the task of constructing intervals via the estimation of quantiles by the name quantile regression too (see e.g. [2,3,5,6]). Especially in the deep learning context, the latter concept has been more prominent with the primary interest in constructing intervals. We will clarify this distinction by highlighting that by “quantile-free regression” we are specifically interested in “quantile-free interval regression”.
>
> When we referred to our method being a direct replacement for quantile regression we had this second, more recent definition of quantile regression in mind. We appreciate the reviewer pointing out that there is some ambiguity in this definition and have therefore updated the phrasing “a direct replacement for quantile regression [...]” to “a direct replacement for quantile regression based interval construction [...]”.
>
> **Action taken**: We will add further distinction in the text between estimating a quantile and estimating an interval. We have also updated the aforementioned phrasing in the abstract.
>
> **References**
>
>
> [1] Zhang, Yufan, Honglin Wen, and Qiuwei Wu. "A contextual bandit approach for value-oriented prediction interval forecasting." IEEE Transactions on Smart Grid (2023).
>
> [2] Chung, Youngseog, et al. "Beyond pinball loss: Quantile methods for calibrated uncertainty quantification." Advances in Neural Information Processing Systems 34 (2021): 10971-10984.
>
> [3] Tagasovska, Natasa, and David Lopez-Paz. "Single-model uncertainties for deep learning." Advances in Neural Information Processing Systems 32 (2019).
>
> [4] Pearce, Tim, et al. "High-quality prediction intervals for deep learning: A distribution-free, ensembled approach." International conference on machine learning. PMLR, 2018.
>
> [5] Fasiolo, Matteo, et al. "Fast calibrated additive quantile regression." Journal of the American Statistical Association 116.535 (2021): 1402-1412.
>
> [6] Maciejowska, Katarzyna, Jakub Nowotarski, and Rafał Weron. "Probabilistic forecasting of electricity spot prices using Factor Quantile Regression Averaging." International Journal of Forecasting 32.3 (2016): 957-965.

---

### Official Review · Reviewer_UxpT · 2023-11-02

**Soundness:** 2 fair
**Presentation:** 3 good
**Contribution:** 2 fair
**Rating:** 5
**Confidence:** 5

**Summary:**

The paper proposes an approach for performing confidence interval regression. To do so, the authors propose a modified version of pinball loss that regresses to the confidence intervals. Since the problem formulation is ill-posed, the papers propose two regularizations on the obtained confidence intervals, namely, interval width and orthogonality. The former promotes shorter intervals, and the latter promotes conditional coverage.

**Strengths:**

- The modified pinball loss formulation for regressing to confidence intervals is interesting.
- Both regularization techniques, minimizing interval width and promoting orthogonality between width and X, make sense to overcome the ill-posedness of the problem.
- The writing is clear, I also appreciated the theoretical results showing the expected coverage based on the modified pinball loss (+regularizations).

**Weaknesses:**

- The quantitative results (Table 3) are not strong enough to justify the need for asymmetric intervals (explained below).
- The evaluation methodology for experiments in Table 3 is potentially flawed (explained in detail below).

**Questions:**

1. Why is it important to report the minimum width interval of a distribution? What practical purpose does it serve to a decision maker who is served with these intervals? The minimum-width intervals come at the cost of losing interpretability on what each end-point constituting the interval means — why is it meaningful to pay this cost? For e.g. in Fig 1a, symmetric intervals convey the following facts: the probability of the random variable to exceed $\mu_h$, probability of the random variable to be lower than $\mu_l$, and that probability of falling within the interval $[\mu_l, \mu_h]$. Whereas in Fig 1b, we only know the probability of falling within $[\mu_l, \mu_h]$. Doesn’t this convey much less information to a decision maker that makes use of these interval estimates?
2. Following the point above, is it possible to know _which_ quantiles do the predicted interval endpoints $\mu_l$ and $\mu_h$ correspond to? This could then partially alleviate the aforementioned limitation.
3. An alternative approach to predicting confidence intervals is to regress to all quantile levels at once, as the authors also noted. Then one can present the smallest interval among all, post-hoc. The authors mention that the limitation of this approach is that SQR [Tagasovska and Lopez-Paz, 2019] performs poorly and thus limits its utility. Recently there have been other approaches [1, 2] to regress all quantiles at once. I encourage the authors to consider evaluating these approaches.
4. *Evaluation methodology.* Firstly, the results suggest that the suggested QFR method results in relatively marginal improvements compared to the trivial baseline QR, and other baselines. This does not fully support the claim of the paper that symmetric intervals are worse than asymmetric ones, as one needs to sacrifice the interpretability of the intervals (see point 1), for very little gain in the interval width. Secondly, and more importantly, I find it concerning that many results in Table 3 were in strikethrough because the coverage requirement is not met. In order to fairly compare two uncertainty estimation methods, it is important that both methods have the same marginal coverage. For example, consider the dataset $\texttt{energy}$, QFR-W has 89.87% coverage and 0.13 width, and QR has 98.31% coverage and 0.18 width, how can one claim that QFR-W is better than QR? One way to carry out a fair comparison is by employing conformal prediction–based wrappers [3] that guarantee marginal coverage, then it is possible to compare different methods in terms of interval width, while they obey the same coverage requirements. Otherwise I am afraid that the comparisons in this table do not fully make sense.

Overall, I found the idea of regressing to the intervals directly via the modified pinball loss clever  and interesting. However, I believe addressing the above points clearly is important for the paper to reach its full potential. I encourage the authors to consider my suggestions.


[1] Brando et al., Deep non-crossing quantiles through the partial derivative, AISTATS 2022.

[2] Rosenberg et al., Fast Nonlinear Vector Quantile Regression, ICLR 2023.

[3] Romano et al., Conformalized Quantile Regression, NeurIPS 2019.

---

> ### Author Response · Authors · 2023-11-15
> **Rebuttal [1/3]**
>
> Thank you for your thoughtful comments and suggestions! We address each of the points raised in turn below.
>
> **_Q1 & Q2 - Interval width vs quantiles as an objective._**
>
> The general view throughout the uncertainty quantification literature is that although several methods might achieve a desired level of marginal coverage with their intervals, this is an underspecified problem with potentially an infinite number of admissible solutions. Therefore, previous works seek to compare solutions based on additional desirable qualities in those intervals. Probably the most popular example of such a quality is interval width which is evaluated ubiquitously (see e.g. the baseline methods to which we compare [1-3]). The idea is simply that the more narrow the region in which we have $\alpha$% confidence that the target lies, the more informative the intervals are for guiding the practitioner on what the final point value will be. The value of narrower intervals for decision-makers has been expressed more formally in previous works such as [4], which explicitly models the tradeoff between the _disutility_ of wider intervals against the _disutility_ of greater miscoverage rates for decision-makers. One example where minimizing interval width is of value is in financial applications where a prediction interval might be compared to some threshold value for a downstream decision (e.g. if a prediction is with $\alpha$ probability below a threshold it is a profitable buy/sell) where more fine-grained prediction intervals can allow for a higher quantity of profitable actions to be taken (see e.g. portfolio selection [5]).
>
> However, we are generally in agreement with the sentiment of this reviewer. Which additional qualities beyond coverage are most desirable is determined by the downstream application. While narrower intervals are a common desiderata, it is indeed true that intervals that represent fixed quantiles may be useful in some cases. The flexibility to provide this decision to the practitioner was a key motivation for this work. Our proposed method optimizes for empirical coverage with the core $\mathcal{L}^{QFR}$ term and allows the user to determine which additional qualities should be optimized for. We proposed two objectives (interval width and conditional coverage) but we also hope that custom, problem-specific regularization terms could also be designed in practice.
>
> One final point we would make is that while the interpretability of quantile-based intervals might sometimes be an aid to decision-makers it can also be harmful. As noted by the reviewer, correct estimates of quantiles would ensure the _count_ of errors to be symmetric on either side of the interval. However, this provides no information on the _magnitude_ of errors which could be disproportionately large on one side for a skewed distribution. Interestingly, previous work in [6] found that providing standard prediction intervals to decision-makers “may have no benefits and may even have negative effects such as reducing their responsiveness to the direction of asymmetric loss”. As there is no one-size-fits-all best solution for decision-makers, we hope that our work better reflects the value of providing practitioners with the ability to optimize predictive intervals for whatever goal is most useful to the final end user.
>
> Regarding Q2 which is an interesting concept: much of the gains obtained from our approach are due to the conditional intervals not being explicitly tied to a fixed quantile. Therefore, this idea is not applicable to our method. However, it would be interesting to consider this as a motivation for an alternative approach to overcoming the limitation of needing to prespecify quantiles in interval regression in future work.
>
> **Action taken**: We have updated the manuscript to better reflect (a) the value of interval width as an objective and (b) the value of providing practitioners with the flexibility to choose whichever interval properties provide the most utility.

---

> ### Author Response · Authors · 2023-11-15
> **Rebuttal [2/3]**
>
> **_Q3 - Related works_**
>
> We thank the reviewer for suggesting these works. Although an interesting paper, Rosenberg et al. extends quantile regression to deal with multivariate outputs (whilst overcoming some of the limitations of previous works on that topic) and is, therefore, addressing a different problem to our work.
>
> Brando et al. propose a method that simultaneously predicts a number of quantile levels (similar to Simultaneous Quantile Regression to which we already compare in our work) with the key innovation of ensuring that these quantiles do not cross (i.e. the 90th quantile would never cross the 85th or 95th quantiles). We evaluate this method as an additional baseline in the setup of our Table 3. As this method is a quantile estimation method and does not prescribe how to construct intervals we include two approaches (a) “narrowest” where we bin search the narrowest interval at each new inference and (b) “symmetric” we use the symmetric quantiles (0.05, 0.95). The results are included below where we report mean interval width (and empirical coverage) on the test set.
>
> | Dataset  |  QFR-W | Narrowest | Symmetric  |
> |---|---|---|---|
> | Concrete  | 0.43 (89.95)  | 0.20 (57.33) | 0.3 (73.30)  |
> | Boston  | 0.51 (89.8)  | 0.26 (64.90)  | 0.36 (78.53)  |
> | Naval  | 0.01 (90.95)  | 0.01 (97.43)  | 0.02 (99.04)  |
> | Energy | 0.13 (89.87)  | 0.07 (50.39)  | 0.14 (72.86)  |
> | Wine  |  0.35 (89.44) |  0.19 (62.56) | 0.28 (76.75)  |
> | Power  |  0.03 (90.73) | 0.02 (73.70)  |  0.03 (89.83) |
> | Protein  | 1.59 (90.07)  | 1.13 (78.04)  | 1.50 (89.26)  |
> | Kin8nm  | 0.34 (90.59)  | 0.25 (70.96)  | 0.38 (85.87)  |
> | Yacht  | 0.27 (93.23)  |  0.29 (51.29) | 0.35 (61.61)  |
>
>
> We note that, as with the previously evaluated methods for estimating all quantiles simultaneously, this approach consistently struggles to maintain coverage at test time. As we discuss in more detail in our discussion on evaluation, we seek solutions that minimize interval width _for a fixed level of coverage_ which is not achieved by this baseline.  We attribute this to two limitations. Firstly, when we select the narrowest possible interval from a set of possible intervals that each obtain marginal coverage, it is not a random choice and it is _more_ likely that we are selecting a particular interval in which coverage is not maintained. Therefore, we would expect that picking the narrowest intervals is likely to result in more miscoverage events than selecting a fixed interval (as is reflected in these results). Secondly, learning all quantiles simultaneously is a more challenging learning problem than simply learning two quantiles for a fixed capacity model. This results in poorer predictive performance at the task in hand.
>
> **Action taken**: We have added these results to the text and extended our discussion on the limitations of estimating all quantiles simultaneously.

---

> > ### Author Response · Authors · 2023-11-15
> > **Rebuttal [3/3]**
> >
> > **_Q4 - Evaluation methodology._**
> >
> > Regarding the improvement in interval width over quantile regression we appreciate that the raw figures may not immediately suggest strong gains but we note that the reduction in interval width is up to 27.7% which is certainly a worthwhile improvement. We also note that our method statistically significantly outperforms quantile regression on 6/9 datasets while tying on the remaining 3. Quantile regression is a very strong baseline and outperforming it by any margin is a result we believe to be useful for practitioners.
> >
> > We agree that the fact that several baselines fail to maintain the desired coverage level on test data to be notable. Although discounting results that fail to achieve coverage is standard in the literature (see e.g. [1]), we also agree that this is something we should have discussed further in the text as it reflects an important dimension on which methods should be assessed. We note that all methods were trained to obtain the same coverage level, therefore failing to achieve the desired coverage level at test time is (a) not something that we can control and (b) reflects a consistent advantage of our proposed method.
> >
> > Regarding the example on the dataset “energy”, we highlight that the goal is to produce the narrowest intervals that contain 90% of test examples which is certainly better achieved by our method (which produces intervals that are 27.7% narrower). Furthermore, while obtaining empirical coverage that is _greater than_ the desired coverage level may often be less harmful than obtaining _less than_ the desired coverage level, at a minimum this still reflects an inefficiency. Additionally, there are many applications in which we are primarily interested in the miscoverage cases (e.g. extreme events) and, therefore, overcoverage may be problematic in addition to being inefficient.
> >
> > Finally, although we agree that conformal prediction is a useful tool for post hoc calibration of intervals, in this work we have restricted ourselves to focus on single-model approaches (see appendix A for our extended discussion on this). We note that because conformal quantile regression is a general-purpose wrapper method for any quantile predictor, research that aims to improve the underlying quantile regression methods is valuable in its own right. Studying the interaction between different quantile regression methods with conformalization is certainly an interesting area for future work.
> >
> > **Action taken**: We have added additional discussion on the joint evaluation of coverage and interval width reflecting this discussion.
> >
> > [1] Tagasovska, Natasa, and David Lopez-Paz. "Single-model uncertainties for deep learning." Advances in Neural Information Processing Systems 32 (2019).
> >
> > [2] Chung, Youngseog, et al. "Beyond pinball loss: Quantile methods for calibrated uncertainty quantification." Advances in Neural Information Processing Systems 34 (2021): 10971-10984.
> >
> > [3] Pearce, Tim, et al. "High-quality prediction intervals for deep learning: A distribution-free, ensembled approach." International conference on machine learning. PMLR, 2018.
> >
> > [4] Landon, Joshua, and Nozer D. Singpurwalla. "Choosing a coverage probability for prediction intervals." The American Statistician 62.2 (2008): 120-124.
> >
> > [5] Huang, Shih-Feng, and Hsiang-Ling Hsu. "Prediction intervals for time series and their applications to portfolio selection." REVSTAT-Statistical Journal 18.1 (2020): 131-151.
> >
> > [6] Goodwin, Paul, Dilek Önkal, and Mary Thomson. "Do forecasts expressed as prediction intervals improve production planning decisions?." European Journal of Operational Research 205.1 (2010): 195-201.

---

### Author Response · Authors · 2023-11-19

Dear Reviewers,

We are extremely grateful for the feedback we have received on our submitted manuscript. The reviews provided several actionable critiques and highlighted a number of important points that could have been made more clearly in our initial draft. We have spent considerable time carefully responding to this feedback and updating the paper with these changes. We believe that this updated version now reflects a much stronger submission addressing each of the reviewers’ stated concerns (we have also provided detailed individual responses in our personal responses on OpenReview below) which we hope will be reflected in a final assessment. Please don't hesitate to let us know if anything in our response requires further clarification or discussion.

**This updated version of the paper is now available to inspect.**

Kind regards,

The Authors

---

### Meta-Review · Area_Chair_qUAA · 2023-12-09

**Metareview:**

The paper tackle the question of set estimation with some theoretical validity guarantee. A classical method to do that is to construct an "confidence" interval for the output and this can be done by estimating the quantiles (by minimizing a pinball loss) of the distribution of the data. The authors argued that such approaches leads to interval centered around the mean which might not be desirable for skewed distribution. The authors then introduce a loss function that allows to directly learn the endpoints of the interval without further restrictions.
This is accompanied by theoretical results showcasing validity of the proposed approach.

The reviewers agree that the introduction of this new loss function for learning prediction interval is quite elegant and have potential impact. However the advantages appears to be less convincing as the non-symmetric interval might be less interpretable and at the end it is task depending to favor this method against classical quantile regression. I believe the authors might want to elaborate more clearly on the advantage of the proposed method.

The theoretical results are also nice but is probably less informative than the authors claims.
Both theorem 3.1 and theorem 3.2 are not providing guarantee on new data points. The first result concern learning with the exact distribution (which I agree validate the method as a valuable heuristic) and the second result, in finite sample, only concern data that was observed. This was mentioned by reviewers but the answer was still not very convincing. I strongly recommend the authors to clearly address this point for future version of the paper. Typically what can be said about $P(y' \in [\mu_1^{\star}, \mu_2^{\star}])$ when $y'$ was not observed during training?


In my opinion, the paper is definitely interesting but the current theoretical and experimental results are not convincing enough. Given the current literature in conformal prediction, that precisely provides interval prediction with finite sample, distribution free guarantee, this paper might need a significant update.

For these reason, I agree with the reviewers and recommend a reject.

**Justification For Why Not Higher Score:**

All the reviewers agree that the current version of the paper is below acceptance threshold.

**Justification For Why Not Lower Score:**

N/A

---

### Decision · Program_Chairs · 2024-01-16

Reject